# Dopamine neurons do not constitute an obligatory stage in the final common path for the evaluation and pursuit of brain stimulation reward

Ivan Trujillo-Pisanty[¤], Kent Conover, Pavel Solis, Daniel Palacios, Peter Shizgal *

Centre for Studies in Behavioural Neurobiology, Concordia University, Montreal, Québec, Canada

¤ Current address: Center for the Neurobiology of Addiction, Pain, and Emotion, Department of Anesthesiology and Pain Medicine, Department of Pharmacology, University of Washington, Seattle, WA, United States of America

* peter.shizgal@concordia.ca

**Data Availability Statement:** Raw data are available from the Concordia University Spectrum

## Abstract

The neurobiological study of reward was launched by the discovery of intracranial self-stimulation (ICSS). Subsequent investigation of this phenomenon provided the initial link between reward-seeking behavior and dopaminergic neurotransmission. We re-evaluated this relationship by psychophysical, pharmacological, optogenetic, and computational means. In rats working for direct, optical activation of midbrain dopamine neurons, we varied the strength and opportunity cost of the stimulation and measured time allocation, the proportion of trial time devoted to reward pursuit. We found that the dependence of time allocation on the strength and cost of stimulation was similar formally to that observed when electrical stimulation of the medial forebrain bundle served as the reward. When the stimulation is strong and cheap, the rats devote almost all their time to reward pursuit; time allocation falls off as stimulation strength is decreased and/or its opportunity cost is increased. A 3D plot of time allocation versus stimulation strength and cost produces a surface resembling the corner of a plateau (the "reward mountain"). We show that dopamine-transporter blockade shifts the mountain along both the strength and cost axes in rats working for optical activation of midbrain dopamine neurons. In contrast, the same drug shifted the mountain uniquely along the opportunity-cost axis when rats worked for electrical MFB stimulation in a prior study. Dopamine neurons are an obligatory stage in the dominant model of ICSS, which positions them at a key nexus in the final common path for reward seeking. This model fails to provide a cogent account for the differential effect of dopamine transporter blockade on the reward mountain. Instead, we propose that midbrain dopamine neurons and neurons with non-dopaminergic, MFB axons constitute parallel limbs of brain-reward circuitry that ultimately converge on the final-common path for the evaluation and pursuit of rewards.

Research Repository (DOI: 10.11573/spectrum.
library.concordia.ca.00986807).

**Funding:** PS; Natural Sciences and Engineering
Research Council of Canada grants #s RGPIN 308-
11,RGPIN-2016-06703; https://www.nserc-crsng.
gc.ca The funders had no role in study design, data
collection and analysis, decision to publish, or
preparation of the manuscript.

**Competing interests:** The authors have declared
that no competing interests exist.

## Introduction

We and our cohabitants are fortunate winners. We have enjoyed reproductive success due to a combination of luck and an array of skills among which acumen in cost/benefit decision making is of paramount importance. Rudimentary ability in this domain can be implemented simply, as is evident from the behavior of animals whose nervous systems comprise only hundreds of neurons [1]. In the multi-million-cell nervous systems of mammals, the foundations of more sophisticated cost/benefit decision making are thought to have been heavily conserved [2, 3]. If so, the rodent species so widely studied in neurobiological laboratories are equipped with variants of decision-making circuitry that continues to shape our own choices and actions.

A seminal moment in the study of the neural foundations of cost/benefit decision making was the discovery that rats would work vigorously and indefatigably for focal electrical stimulation of sites in the basal forebrain and midbrain [4]. Despite the artificial spatiotemporal distribution of the evoked neural activity, the rats behaved as if procuring a highly valuable, natural goal object, such as energy-rich food. This striking phenomenon, dubbed "intracranial self-stimulation" (ICSS), has been investigated subsequently by means of perturbational, pharmacological, correlational, computational, and behavioral methods that have seen dramatic recent improvements in precision, specificity, and power. For example, the electrical stimulation employed originally activates neurons near the electrode tip with relatively little specificity, whereas contemporary optogenetic methods restrict activation to genetically defined neural populations. Currently employed pharmacological agents are far more selective than the drugs employed in the early studies. Whereas the performance function linking response vigor to reward magnitude is inherently non-linear and is subject to distortion by disruptive side-effects of drugs and motoric activation, contemporary psychophysical methods "see through" this function so as to support inferences about the value of the induced reward as well as the form and parameters of the functions that map observable inputs and outputs into the variables that determine behavioral-allocation decisions. In the current study, we combine, for the first time, direct, specific optical activation of midbrain dopamine neurons, modulation of dopaminergic neurotransmission by a highly selective dopamine-reuptake blocker, psychophysical inference of the values of variables underlying cost-benefit decision making, computational modeling of the processes that intervene between the optical activation of the dopamine neurons and the consequent behavioral output, and simulation of model output. The results challenge a long-dominant model in which dopaminergic neurons constitute an obligatory stage in the final common path for the rewarding effect manifested in ICSS.

### The role of midbrain dopamine neurons in ICSS

Performance for rewarding brain stimulation has long been known to depend on dopaminergic neurotransmission. Drugs that boost dopamine signaling decrease the strength of the stimulation required to support a given level of ICSS, whereas drugs that decrease dopamine signaling necessitate a compensatory increase in stimulation strength at lower doses and eliminate responding at higher ones [5, 6]. It was believed initially that these effects are due to direct activation of dopaminergic neurons by the electrical stimulation. However, midbrain dopaminergic fibers are very fine and unmyelinated [7]. Consequently, they have very high thresholds to excitation by extracellular currents. Robust ICSS of sites along the medial forebrain bundle (MFB) is observed using stimulation parameters too weak to produce substantial recruitment of dopaminergic fibers [8]. Moreover, estimates of recovery of refractoriness, conduction velocity, and frequency-following fidelity in the directly activated fibers underlying the rewarding effect implicate fibers that are myelinated and far more excitable than those of

dopamine neurons [9–14]. To reconcile these observations with the pharmacological evidence for dopaminergic modulation of ICSS, a "series-circuit" model was proposed [10–12, 15, 16]. According to this model, myelinated fibers of non-dopaminergic neurons transsynaptically activate midbrain dopamine neurons, thus generating the rewarding effect. This model of brain reward circuitry, which has remained virtually unchallenged for nearly forty years, fails to provide a cogent account for the new data and simulations we report here. We thus propose a fundamental revision in which the phasic firing of dopaminergic neurons constitutes only one of the multiple signals that converge on the final common path subserving the evaluation and pursuit of rewards.

## The roots of the current study

The roots of the current experiment on the rewarding effect produced by optical activation of midbrain dopamine neurons lie in earlier work in which electrical stimulation of the MFB served as the reward. We use the acronyms, eICSS and oICSS, to refer to operant performance for electrical and optical brain stimulation, respectively (Table 1). Four lines of work on eICSS gave rise to the current oICSS study:

1. characterization of spatiotemporal integration in the underlying neural circuitry,

2. measurement of how the intensity of the rewarding effect grows as a function of the aggregate rate of induced firing in the directly activated neurons,

3. measurement and modeling of how performance for the electrical reward depends on its strength and cost, and

4. determination of the stage of processing at which perturbation of dopaminergic neurotransmission alters eICSS.

**The counter model.** According to the "counter model" of spatiotemporal integration in the neural circuitry underlying eICSS [17–19], the neural signal that gives rise to the rewarding effect reflects the aggregate rate of induced firing produced by a pulse train of a given duration. Neither the number of activated neurons nor the rate at which they fire matter per se; it is their product that determines the intensity of the rewarding effect. The counter model is well supported empirically [20–23]. An analogous coding principle has been proposed by Murasugi, Salzman & Newsome [24] to account for the effect of electrical microstimulation of cortical area V5 on visual-motion perception.

**The growth of reward intensity as a function of the aggregate rate of firing.** An analogy may help convey what we mean by "reward intensity." Imagine that a rat tastes a sucrose solution. The rat is thought to derive two different kinds of information from this input: a) sensory data indicating the concentration and identity of the tastant and b) evaluative data indicating what the sucrose is worth to the rat in its current physiological and ecological state [25, 26]. These two signals could diverge, for example when an overfed rat encounters increasingly concentrated solutions. In that case, the sensory "sweetness" signal may continue to increase while the evaluative "goodness" signal plateaus or declines. We use the term, "reward intensity," by analogy to the evaluative signal. Indeed, we have shown that rats can compare the reward-intensity signals produced by MFB stimulation and intraoral sucrose so as to determine which is larger and can combine them such that a compound electrical-gustatory reward is worth more than either of its constituents delivered singly [27].

The reward-growth function translates the aggregate rate of stimulation-induced firing into the intensity of the rewarding effect. Gallistel's team used operant matching to describe this function [21–23, 28]. According to the matching law [29–32], subjects partition their time

**Table 1. Definition of acronyms and symbols.**

| Acronym or Symbol | Definition |
|---|---|
| $a$ | price-sensitivity exponent |
| AIC | Akaike Information Criterion |
| BSR | brain stimulation reward |
| $C_r$ | Conditioned reward value |
| DAPI | 4′,6-diamidino-2-phenylindole |
| eICSS | electrical intracranial self-stimulation |
| eYFP | enhanced yellow fluorescent protein |
| $F_{firing}$ | firing frequency induced by electrical or optical stimulation |
| $F_{firing_{hm}}$ | firing frequency that generates a reward of half-maximal intensity |
| $F_{pulse_{hm}}$ | pulse frequency required to drive reward intensity to half its maximum value |
| $F_{pulse_{hm}}^*$ | estimated pulse frequency required to drive reward intensity to half its maximum value if frequency-following fidelity were perfect |
| $g$ | exponent governing the steepness of reward-intensity growth |
| ICSS | intracranial self-stimulation |
| $K_{rg}$ | reward-intensity scalar |
| MFB | medial forebrain bundle |
| oICSS | optical intracranial self-stimulation |
| $P_{obj_e}$ | objective price at which time allocation is halfway between its minimum and maximum values |
| $P_{obj_e}^*$ | estimated objective price at which time allocation is halfway between its minimum and maximum values when frequency-following fidelity is perfect |
| $R_{bsr}$ | peak reward intensity achieved over the course of a stimulation train |
| $T_{max}$ | maximal time allocation |
| $T_{min}$ | minimal time allocation |
| TH | tyrosine hydroxylase |
| YFP | shorthand for eYFP |

between two concurrent variable-interval schedules in proportion to the relative payoffs from the two schedules. If so, when two schedules are each configured to deliver stimulation trains, the relative payoffs can be inferred from the ratios of work times and reward rates.

Simmons and Gallistel [23] brought out a key feature of the reward-growth function: given a sufficiently high current, reward intensity saturates at pulse frequencies well within the frequency-following capabilities [8] of the directly stimulated substrate. In other words, reward intensity levels off as pulse frequency increases even though the output of the directly stimulated neurons continues to grow. The reward-growth function they described is well fit by a logistic [33], a function that is shaped like an inverted hockey stick (Fig 1) when plotted on double logarithmic coordinates. As we will show, it is the non-linear form of this growth function that prevents the series-circuit model from explaining the differential effects of dopamine-transporter blockade on eICSS and oICSS. To account for the oICSS data, the contribution of dopamine neurons must be brought to bear on the input side of the reward-growth function as well as on the output side. To account for the eICSS data [34], dopamine neurons must intervene on the output side but not on the input side.

**The reward-mountain model.** The behavioral method employed in this study entails measurement of time-allocation decisions [35] by laboratory rats. The method is based on a model (Fig 2) of how time allocation is determined by the strength and cost of reward. According to Herrnstein's single-operant matching law [30, 31, 36], subjects performing an operant

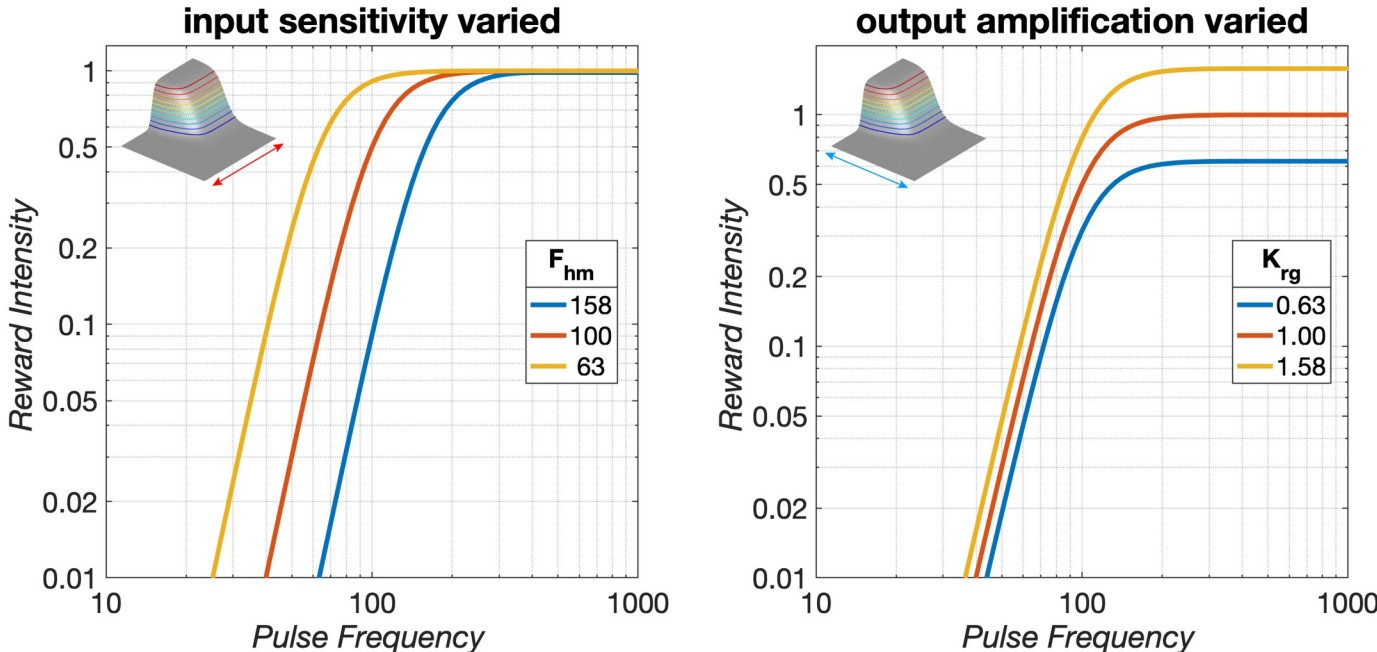

**Fig 1. The reward-growth function.** Gallistel and colleagues have shown that the pulse frequency is translated into the intensity of the rewarding effect by a saturating function [21–23, 28], such as a logistic [33] (Eq S11). We assume that in those experiments, the induced firing frequency can be equated to the pulse frequency [8]. The position of the growth curve along the X axis is determined by the pulse frequency that drives reward intensity to half its maximal value ($F_{hm}$, also called $F_{pulse_{hm}}$ below), whereas the position along the Y axis (e.g., the maximal reward intensity attained) is determined by a parameter we call $K_{rg}$. In the left panel, $F_{hm}$ is varied while $K_{rg}$ is held constant, whereas in the right panel, $K_{rg}$ is varied while $F_{hm}$ is held constant. The inserts are described below, after the reward mountain has been introduced. The pulse frequencies shown here are typical of eICSS experiments.

response, such as lever pressing, to obtain an experimenter-controlled reward partition their time between "work" (performance of the response required to obtain the reward), and "leisure" (performance of alternate activities such as grooming, exploring, and resting). The higher the benefit from the experimenter-controlled reward and the lower its cost, the larger the proportion of time devoted to work.

The reward-mountain model [37–39] treats the rewarding stimulation as a fictive benefit; although it satisfies no known physiological need, the effect of the stimulation mimics goal objects that do [15, 27]. Two types of costs are incorporated in the model. The first is the intensity of the perceived effort entailed in meeting the response requirement, which consists of holding down a lever for a duration determined by the experimenter. While the rat works to hold down the lever, it cannot groom, rest, or explore. Thus, it pays an opportunity cost [40], which consists of the benefits that would have been obtained from the foregone alternative activities.

In the spirit of the expanded matching law [32], the reward-mountain model equates the payoff from work to the ratio of its benefits and costs. A behavioral-allocation function [38] derived from the generalized matching law [41] compares the payoffs from work and leisure so as to determine the allocation of time to these two sets of activities.

**The effect of perturbing dopamine neurotransmission on the reward mountain.** The curve-shift [42–44] or progressive-ratio [45] methods are typically regarded as the "gold standards" for measuring drug-induced changes in the behavioral effectiveness of brain-stimulation reward. These methods assess shifts in the functions relating performance vigor to electrical pulse frequency (in the case of the curve-shift method) or to the number of responses

# Core components of the mountain model

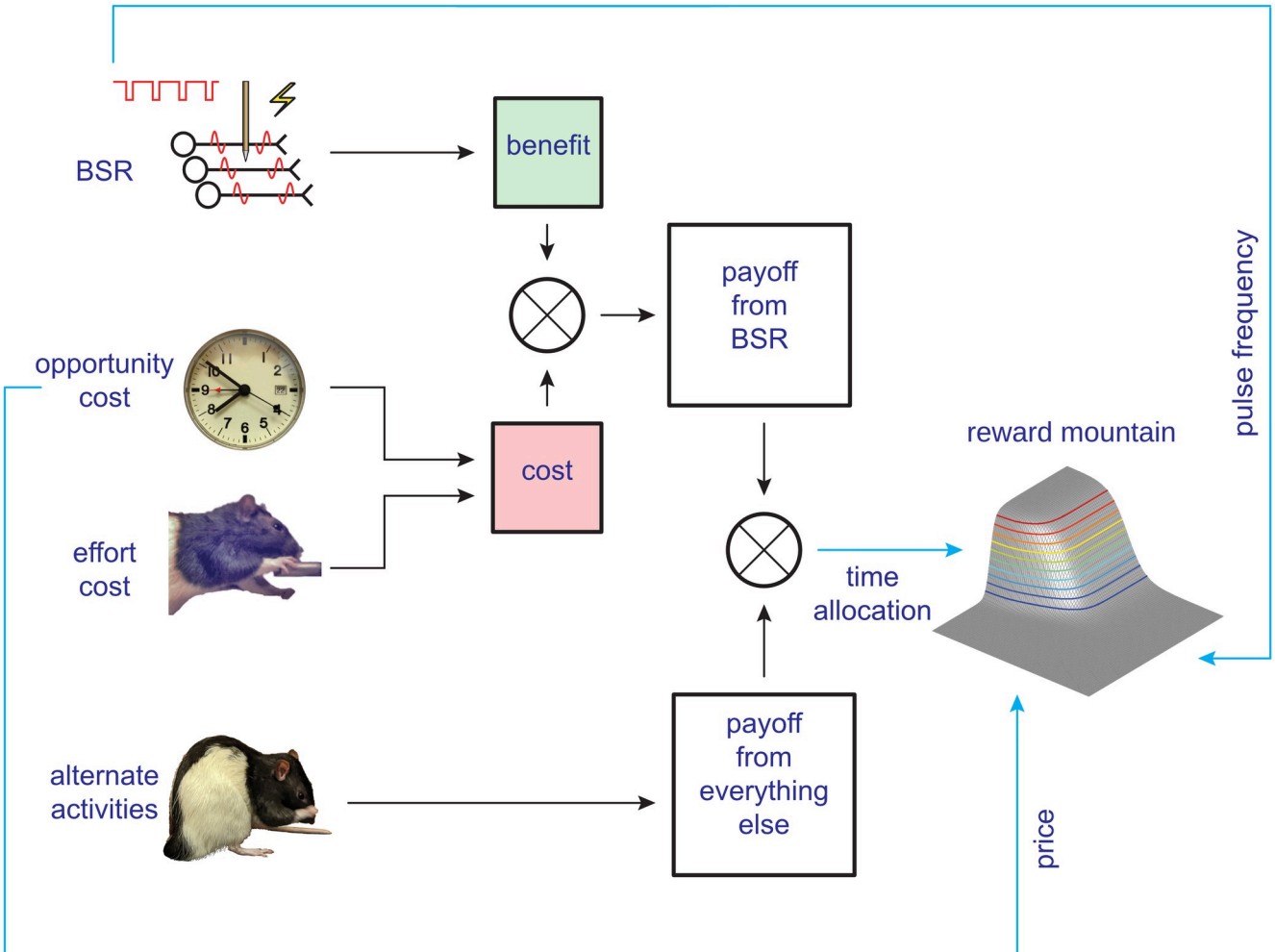

**Fig 2. Core components of the reward-mountain model.** The self-stimulating rat partitions its time between working for the rewarding stimulation and performing alternate activities, such as grooming, exploring, and resting. The payoff from work depends on the benefit it provides and the cost it entails. The benefit arises from the induced neural activity (shown here to arise from electrical stimulation), whereas the costs are of two different sorts: the intensity of the perceived effort entailed to meet the response requirement and the opportunity cost of the time so expended. The ratio of benefits to costs constitutes the payoff from the experimenter-controlled reward, which is compared to the payoff from alternative activities by means of a behavioral allocation function derived from Herrnstein's single-operant matching law. The result of this comparison determines allocation of the subject's time.

required to earn a reward (in the case of the progressive ratio method). The reward-mountain model shows that these two-dimensional methods yield fundamentally ambiguous results [37, 39]. Performance depends both on the strength of the electrical reward (determined by the pulse frequency) and on response cost (determined, in progressive-ratio testing, by the number of required responses per reward). This dependence is described by a surface in a three-dimensional space (reward-seeking performance versus pulse frequency and response cost (Fig 2)). When the surface is shifted along one of the axes representing the independent variables, its silhouette may also shift along the orthogonal axis [37–39]. An observer using either the curve-shift or progressive-ratio methods views only the silhouette of the surface and thus cannot determine in which way the surface itself (rather than its silhouette) has been displaced.

Did administration of a drug shift the reward-growth function along the pulse-frequency axis, displace the mountain along the cost axis, or both? An observer using either of these convention methods cannot know. In contrast, an observer using the three-dimensional reward-mountain model can answer definitively because the direction of displacement is determined unambiguously [37–39].

Fig 1 shows the content of the green box in Fig 2 labeled "benefit:" the reward-growth function. Shifts of this function along the pulse-frequency axis (left panel) reflect changes in *input* sensitivity. These changes alter the pulse frequency required to drive reward intensity to a given proportion of its maximum. This is tantamount to rescaling the *input*. When input sensitivity changes, the reward-growth function shifts laterally, dragging the reward mountain with it along the pulse-frequency axis (see insert). In contrast, one of the factors that can shift the reward mountain along the cost axis (right panel) is a change in *output* amplification (called "gain" in our previous papers [34, 39, 46, 47]). This alters the maximal reward intensity without changing the pulse frequency required to achieve this maximum (or any other proportion of the maximal intensity); the reward-growth function is shifted vertically along the logarithmic axis representing reward intensity. Because all non-zero reward intensities have now been boosted or cut, willingness to pay for them changes accordingly, and the mountain shifts along the cost axis (see insert). However, that shift is not unique. For example, due to the scalar combination of benefits and costs (as represented by the circle containing an X that combines benefits and costs in Fig 2), indistinguishable shifts along the cost axis result from multiplying the benefits by a constant or from dividing the costs by the same constant. Thus, the reward-mountain method does not unambiguously distinguish changes in benefits and costs. What it does instead is to distinguish changes in input sensitivity (left panel of Fig 1) from all the other determinants of reward valuation.

Shizgal's team has used the reward-mountain model to assess the effects of perturbing dopaminergic neurotransmission on performance for rewarding electrical stimulation of the MFB [34, 39, 47]. They found that enhancement of dopaminergic neurotransmission by means of dopamine/transporter blockade [34, 39] or attenuation by means of dopamine-receptor blockade [47] shift the reward mountain almost uniquely along the axis representing response cost. This implies that, contrary to what was long believed [48, 49], the contribution of dopaminergic neurotransmission to eICSS is brought to bear downstream (at, or beyond, the output) of the reward-growth function, either by rescaling this output, the costs that are combined with it, or the value of alternate activities ("everything else" in Fig 2).

We show here that one of the contributions of dopaminergic neurons to *oICSS* is brought to bear at, or prior to, the input to the reward-growth function; it alters the input sensitivity of the reward-growth function. This contrasts sharply with the case of *eICSS* in which such changes in input sensitivity have rarely been seen following perturbation of dopaminergic neurotransmission in reward-mountain experiments [34, 39, 47].

## The empirical question and its significance

Rats [50] and mice [51, 52] will perform operant responses to obtain direct optical stimulation of midbrain dopamine neurons. The specific empirical question posed in the present study is whether the effect of dopamine-reuptake blockade on performance for rewarding optical stimulation of midbrain dopamine neurons, as measured by shifts in the reward mountain, mimics the effect of this manipulation on performance for rewarding electrical stimulation of the MFB [34, 39]. The outcome matters in at least two ways. First, it bears on the issue of how midbrain dopamine neurons contribute to reward evaluation and pursuit. Second, it bears on and challenges the series-circuit model that, on superficial consideration, might appear to integrate the

**Fig 3. The series-circuit model.** The boxed low-pass filter symbols represent the frequency-following functions that map the electrical pulse-frequency into the induced frequency of firing in the directly stimulated neurons subserving eICSS or the optical pulse-frequency into the induced frequency of firing in the dopamine neurons subserving oICSS. The reward-growth functions are shown here as S-shaped, which is the form they describe when plotted in linear or semi-logarithmic (reward intensity versus $log$(pulse frequency)) coordinates. $K_{ds}$ ($ds$ stands for "directly stimulated") scales the input to the reward-growth function for eICSS, whereas $K_{da}$ scales the input to the reward-growth function for oICSS. $K_{rg}$ scales the output of the reward-growth functions. According to this model, the rewarding effects produced by both electrical stimulation of medial-forebrain-bundle (MFB) neurons and optical stimulation of midbrain dopamine neurons have a common cause: activation of midbrain dopamine neurons. This activation is due to transsynaptic excitation in the case of eICSS and direct excitation in the case of oICSS. The results of prior eICSS studies [34, 39, 47] require that drugs that perturb dopaminergic neurotransmission alter the computation of reward intensity by actions downstream from the output of a logistic reward-growth function (leftmost box containing an S-shaped curve). The results of the present study require that a similar reward-growth function lies downstream from the dopamine neurons (rightmost box containing an S-shaped curve). We show below that the placement of this second reward-growth function is incompatible with the eICSS data.

findings obtained by means of direct, specific, optical activation of dopamine neurons with the older findings obtained by means of electrical stimulation that activates dopamine neurons indirectly.

Fig 3 illustrates the series-circuit model [10–12, 15, 16], which holds that the rewarding effect produced by electrical stimulation of the MFB arises from the transsynaptic activation of midbrain dopamine neurons. The left half of the figure depicts the generation of the reward-intensity signal subserving eICSS. The observed displacement of the reward mountain by perturbation of dopaminergic neurotransmission [34, 39, 47] requires that the dopaminergic neurons lie downstream from the output of the reward-growth function for eICSS.

The right half of Fig 3 depicts the generation of the reward-intensity signal subserving oICSS. Dopamine-transporter blockade increases stimulation-induced dopamine release and thus rescales the ***input*** to the reward-growth function shown in the right-hand portion of Fig 3 (via the triangular amplifier symbol labeled "$K_{da}$"). Such an effect shifts the reward-growth function leftward along the logarithmic pulse-frequency axis due to the increased "bang for the buck," dragging the reward mountain with it (Fig 1)). The results confirm this prediction. These shifts along the pulse-frequency axis are orthogonal to those typically observed in the eICSS studies [34, 39, 47]. We show below that the series-circuit model founders on this discrepancy: it fails to provide a cogent, unified account of both the prior eICSS and present oICSS data. Thus, we advocate abandoning the series-circuit model that has figured so heavily in accounts of eICSS over the past four decades. In its place, we develop a new account entailing parallel dopaminergic and non-dopaminergic pathways that ultimately converge on a final common path. That model can accommodate both the prior eICSS and the present oICSS findings.

## The issues addressed by the simulations

Multiple non-linear functions intervene between the stimulation we deliver to the brain and the behavioral consequences we observe.

1. The firing frequency of the activated neurons eventually rolls off as pulse frequency is increased sufficiently [8].

2. The intensity of the rewarding effect is a non-linear, saturating function of the aggregate rate of induced impulse flow [22, 23, 28].

3. Subjective opportunity cost is a non-linear function of objective opportunity cost [40].

4. Although the form of the function has yet to be described empirically, subjective effort cost is almost surely a non-linear function of the objective work requirement, rising towards infinity as the physical capabilities of the subject are exceeded.

5. Allocation of behavior to eICSS or oICSS is an increasing sigmoidal function of reward intensity and a decreasing sigmoidal function of subjective cost [30, 31, 37–39].

Intuitive analysis and verbal reasoning rapidly come to grief when confronted with multiple, interconnected non-linearities. To understand how such a set of non-linear, interacting functions produces systematic behavior, it is necessary to capture the known relationships in a quantitative model and to simulate its output in response to the experimental inputs. This is what we have done to complement the empirical experiment. These simulations are summarized here and reported in detail in the accompanying Matlab® Live Script.

The modeling and simulations provide mathematical and logical support for the proposed interpretation of the experimental results and their integration into a new, unified account of both oICSS and eICSS. They provide a systematic framework for working out the minimal set of "moving parts" that determine reward evaluation and pursuit, making testable predictions about the effects of future manipulations, rethinking the role of dopamine neurons, and guiding efforts to identify the non-dopaminergic components of the neural circuitry in which the dopamine neurons are embedded.

## Results

Seven rats completed both phases of the study: the power-frequency trade-off and the pharmacological experiment. The data are presented first in two dimensions. The reward-mountain model is then applied to integrate the results of the pharmacological experiment into a three-dimensional structure, to document the effects of dopamine/transporter blockade on the position of the reward mountain, and to interpret these displacements in terms of drug-induced changes in the values of the variables that determine reward pursuit.

### Power-frequency trade-off

The data in Fig 4 were obtained using the FR-1 task. The number of responses emitted per 2-min trial by an exemplar rat (Bechr29) is shown as a function of pulse frequency and optical power; the results for the remaining rats are shown in the S1 File (S1-S6 Figs). Response rates grew as the pulse frequency was increased, in some cases (Bechr26-29) up to, or well beyond, optical pulse frequencies of 40 pulses s$^{-1}$. In rats Bechr14, 19, 21, and 27, maximum response rats grew as a function of pulse frequency over the lower powers but approached asymptote as the power was further increased. In the remaining rats, maximum response rates grew as a function of pulse frequency over the entire range of tested powers. The position of the rate-frequency curves generally shifted leftwards as optical power was increased.

### Time allocation as a function of reward strength and cost

The data in Fig 5 were obtained using the lever-hold-down task. The two-dimensional views shown here were eventually combined (see below) to generate three-dimensional reward mountains. The curves in Fig 5 plot time allocation as a function of reward strength ("Pulse Frequency") or opportunity cost ("Price"). The results are from an exemplar rat (Bechr29). Panel **A**

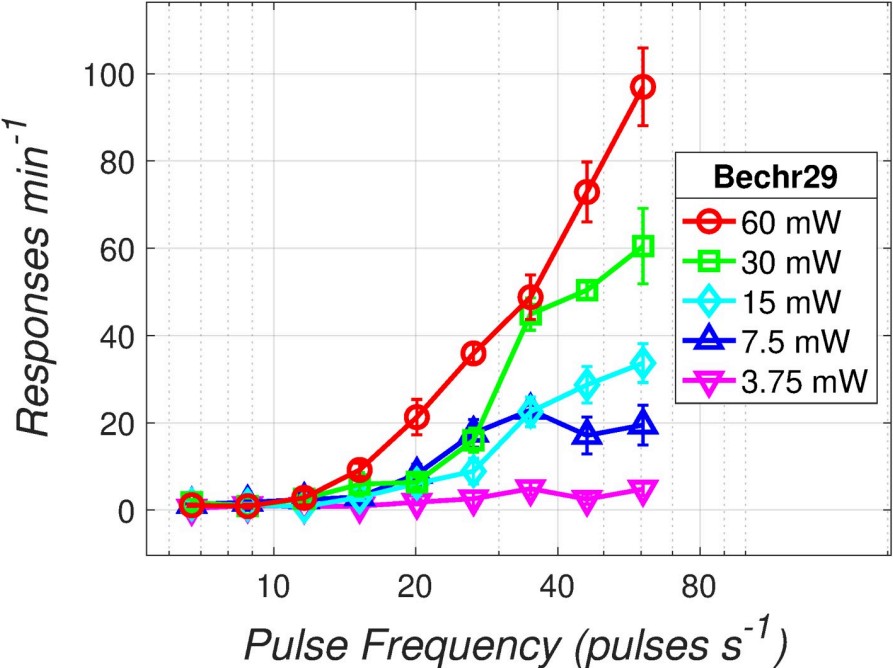

**Fig 4. Rate-frequency curves as a function of optical power.** The number of responses emitted per 2-min trial by an exemplar rat (Bechr29) is plotted as a function of pulse frequency and optical power.

shows frequency-sweep results: Time allocation increased as a function of optical pulse frequency. The time/allocation/versus/pulse/frequency curve obtained following blockade of the dopamine transporter by GBR-12909 is shifted leftwards with respect to the curve obtained in the vehicle condition. Panel **B** shows price-sweep results: Time allocation decreased as a function of increases in opportunity cost (cumulative time that the lever had to be held down to trigger a reward). The time/allocation/versus/price curve obtained following blockade of the dopamine transporter is shifted rightwards with respect to the curve obtained in the vehicle condition. Radial-sweep data were obtained by conjointly decreasing the pulse frequency and increasing the price in stepwise fashion over consecutive trials. Panels **C** and **D** show the same radial-sweep results from two orthogonal vantage points. Time allocation is plotted against pulse frequency in panel **C** and against price in panel **D**. Time allocation increased as a function of pulse frequency and decreased as a function of price. The time-allocation curve obtained following blockade of the dopamine transporter is shifted leftwards along the pulse-frequency axis with respect to the curve obtained in the vehicle condition and rightwards along the price axis. Graphs for the remaining rats are shown in the S1 File (S7-S12 Figs).

The drug-induced shifts in the curves shown in Fig 5 and S7-S12 Figs are inherently ambiguous: displacement of these curves along a given axis could be due to any combination of displacements of the underlying reward-mountain structure along the price and/or pulse-frequency axes [38, 39, 53]. This ambiguity is removed by fitting the reward-mountain model, which expresses time allocation as a function of both the price and strength of the rewarding stimulation. In the three-dimensional space of the reward-mountain model, we can determine unambiguously the degree to which dopamine-transporter blockade displaces the mountain along the price and pulse-frequency axes. By so doing, we distinguish the effect of dopamine-transporter blockade at, or prior to, the input to the reward-growth function (Fig 1) from the effect at, or beyond, the output of this function. As we will show, the fate of the series-circuit

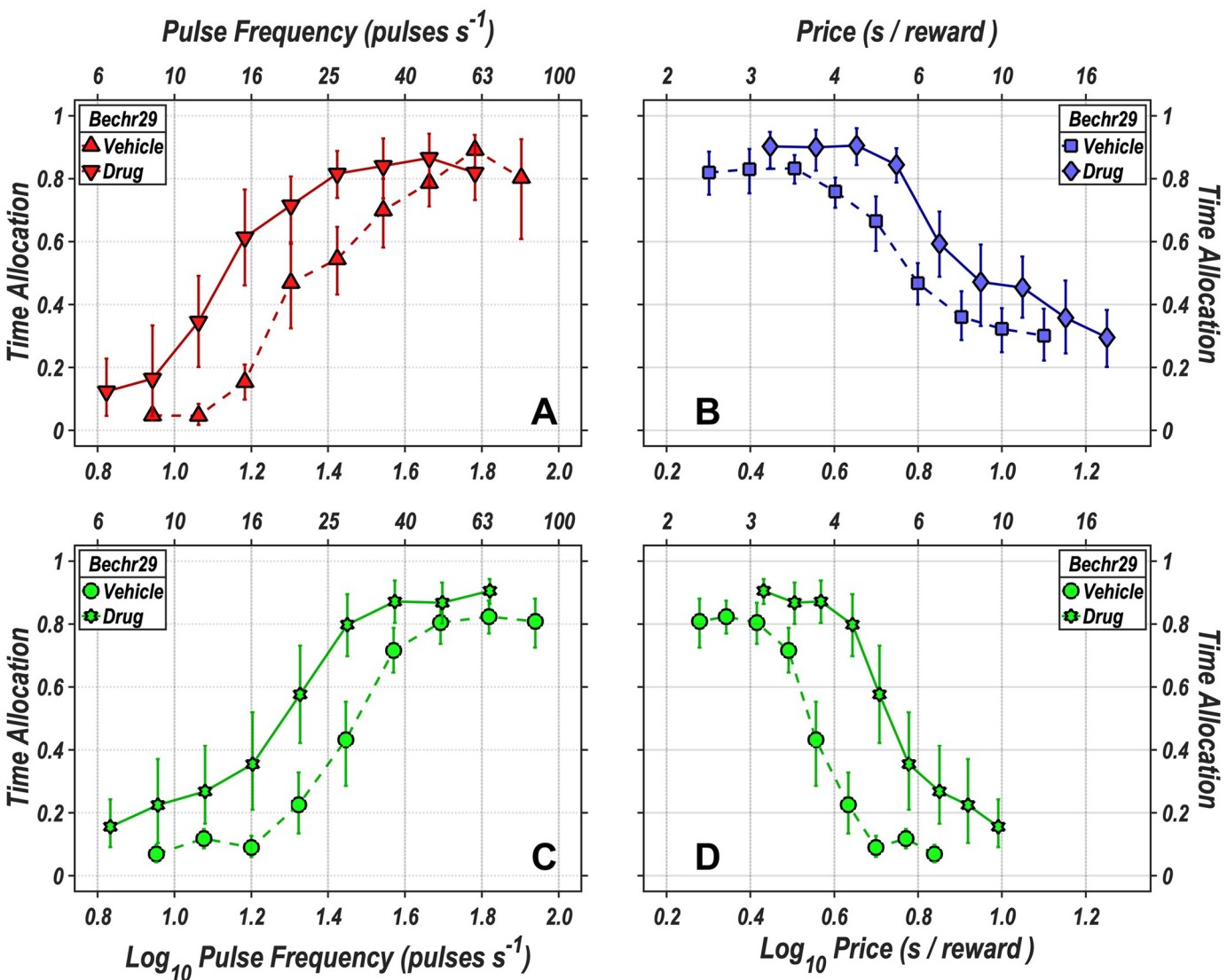

**Fig 5. Time allocation as a function of reward strength and cost. A**: Time allocation as a function of pulse frequency (reward strength) in the vehicle (upright triangles) and drug (inverted triangles) conditions. **B**: Time allocation as a function of price (opportunity cost) in the vehicle (squares) and drug (diamonds) conditions. In the radial-sweep condition, the pulse frequency was decreased and the price decreased concurrently, in stepwise fashion, over consecutive trials. Time allocation is plotted as a function of pulse frequency in panel **C**: and as a function of price in panel **D**: Data from the vehicle condition are represented by circles, whereas data from the drug condition are represented by Stars of David. The error bars represent 95% confidence intervals. Data are from an exemplar rat (Bechr29).

model hangs on that distinction and founders on the fact that specific dopamine-transporter blockade by GBR-12909 shifts the reward mountain for oICSS (present data), but not eICSS [34], along the pulse-frequency axis.

## Model fitting and selection

The standard version of the reward-mountain model has six parameters: $\{a, F_{pulse_{hm}}, g, P_{obj_e}, T_{max}, T_{min}\}$. The $F_{pulse_{hm}}$ and $P_{obj_e}$ parameters set the location of the mountain along the pulse-frequency and price axes, respectively. $F_{pulse_{hm}}$ (shortened to $F_{hm}$ in the graph legends) is the pulse frequency at which reward intensity is half maximal, whereas $P_{obj_e}$

is the price at which time allocation to pursuit of a maximal reward falls midway between its minimal and maximal values, $T_{min}$ (minimum time allocation) and $T_{max}$ (maximal time allocation). The slope of the mountain surface along the price axis is set by the price-sensitivity exponent, $a$, whereas the slope along the pulse-frequency axis is determined both by $a$ and by the reward-growth exponent, $g$.

In previous studies ([34, 39]), a seven-parameter version of the reward-mountain model sometimes performed better than the standard six-parameter version. The added parameter accommodates cases in which minimum time allocation is higher at low prices than at higher ones. This parameter is called $C_r$ and has been interpreted to represent a reward value assigned to the lever and/or to the act of pressing it [39].

Both the six- and seven-parameter versions of the reward-mountain model are derived in the S1 File. (See: Derivation of the reward-mountain model).

There is a trade-off between the number of parameters fit to a dataset and the precision with which the value of each parameter can be estimated. In order to restrict the number of fitted parameters, we forced common values of $T_{max}$ and $T_{min}$ to be fit to the vehicle and drug data. The two location parameters were always free to vary across the vehicle and drug conditions. Four variants of the six-parameter model were fit. The variants are distinguished by whether either, both, or neither of the $a$ and $g$ parameters were free to vary across the vehicle and drug condition. Eight variants of the seven-parameter model were fit. These variants are distinguished by the combinations of the $a$, $C_r$, and $g$ parameters that were free to vary across the vehicle and drug conditions.

The 12 candidate models (four variants of the six-parameter version and eight variants of the seven-parameter version) are described in S3 Table. The Akaike Information Criterion (AIC) [54] was used to identify the best-fitting model for each rat. This statistic implements a trade-off between goodness of fit and simplicity. Thus, the AIC penalizes models with large numbers of parameters in comparison to simpler ones. The fits of all of the candidate models to the reward-mountain data from all seven rats converged successfully. Detailed results for rat Bechr29 are shown in S4 Table, and results for all rats are shown in S5,S6 Tables.

## Fitted reward-mountain surfaces

The reward-mountain surfaces fit to the vehicle and drug data from rat Bechr29 are shown in Fig 6, whereas those fit to the data from the remaining rats are shown in the S1 File (S17-S22 Figs).

To facilitate visualization of the drug-induced shift in the location of the reward mountain in Fig 6, the surfaces have been re-plotted as contour graphs in Fig 7. Horizontal comparison in Fig 7 shows that blockade of the dopamine transporter shifted the mountain downwards along the pulse-frequency axis, whereas vertical comparison shows that the mountain shifted rightwards along the price axis. The shifts are summarized in the bar graph; dot-dash cyan lines show estimates corrected for changes in frequency-following fidelity due to the drug-induced displacement of the mountain along the pulse-frequency axis. The rationale for the correction and the details of its implementation are described in the S1 File. (See sections *Parameters of the frequency-following function for oICSS* and *Correction of the location-parameter estimates for changes in frequency-following fidelity*). Contour and bar graphs for the remaining rats are shown in the S1 File (S23-S28 Figs).

## Location-parameter estimates

**F$_{hm}$:** The surface-fitting procedure returns the location-parameter value, $F_{pulse_{hm}}$, that positions the reward mountain along the ***pulse***-frequency axis. In studies of eICSS employing the reward-mountain model, it was assumed that this value corresponded to the induced ***firing***

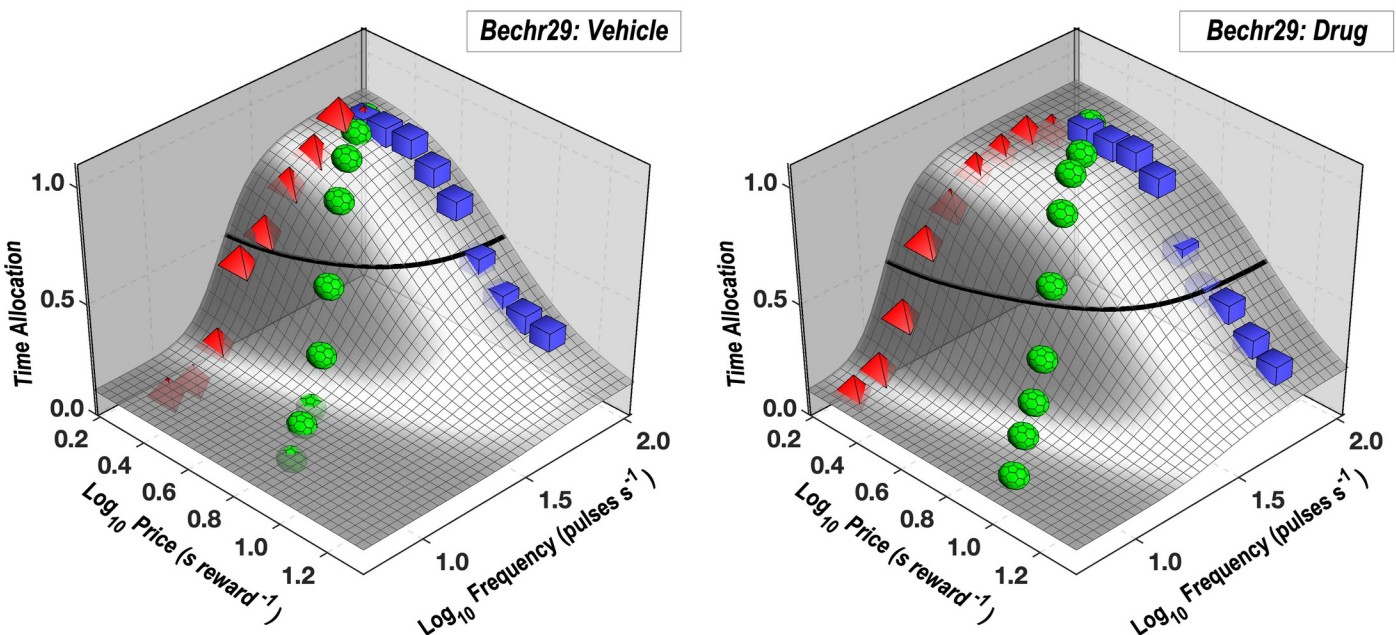

**Fig 6. Reward-mountain surfaces fit to the vehicle and drug data from rat Bechr29.** The surfaces of the reward-mountain shell are shown in gray. The thick black line represents the contour mid-way between the minimal and maximal estimates of time allocation (the estimated altitudes of the valley floor and summit). Mean time-allocation values for the pulse frequency, price, and radial sweeps are denoted by red pyramids, blue squares, and green polyhedrons, respectively.

frequency in the directly activated neurons. This firing frequency is the location parameter of the underlying reward-growth function. Given the exceptional frequency-following fidelity of the directly activated neurons subserving eICSS of the MFB [8], it is not unreasonable to assume that each pulse elicits an action potential in most or all of the directly stimulated MFB neurons when the pulse frequency equals $F_{pulse_{hm}}$. In contrast, the findings reviewed in section *Parameters of the frequency-following function for oICSS* of the S1 File argue that such an assumption is untenable in the case of oICSS of channelrhodopsin-2 expressing midbrain dopamine neurons. We therefore used the data from the rate-frequency curves obtained here (Fig 4 and S1-S6 Figs) and prior studies (e.g., [55]) to estimate the firing frequencies corresponding to the $F_{pulse_{hm}}$ values. We label these as $F^*_{pulse_{hm}}$ (rather than as $F_{firing_{hm}}$) because these values will be plotted along the ***pulse***-frequency axis, and we refer to them as "corrected" estimates of $F_{pulse_{hm}}$.

Table 2 shows the estimated drug-induced ***shifts*** in the location of the reward-mountain core along the pulse-frequency axis. The shifts are the differences between the common logarithms of the estimated firing frequencies ($F^*_{pulse_{hm}}$) that produced half-maximal reward intensities in the drug and vehicle conditions (S7 Table). Also included are the differences between the estimates for the drug and vehicle conditions and the 95% confidence intervals surrounding these differences. In all cases, the confidence band excludes zero, thus meeting our criterion for a statistically reliable effect. In six of seven cases, the sign of the difference is negative, indicating that the drug shifted the reward mountain downwards along the pulse-frequency axis. Note that the one discrepant shift (for Rat Bechr27) is the smallest. The values in the "$K_{da_{drug}}$" column give the proportional reduction in $F^*_{pulse_{hm}}$ produced by dopamine-transporter blockade.

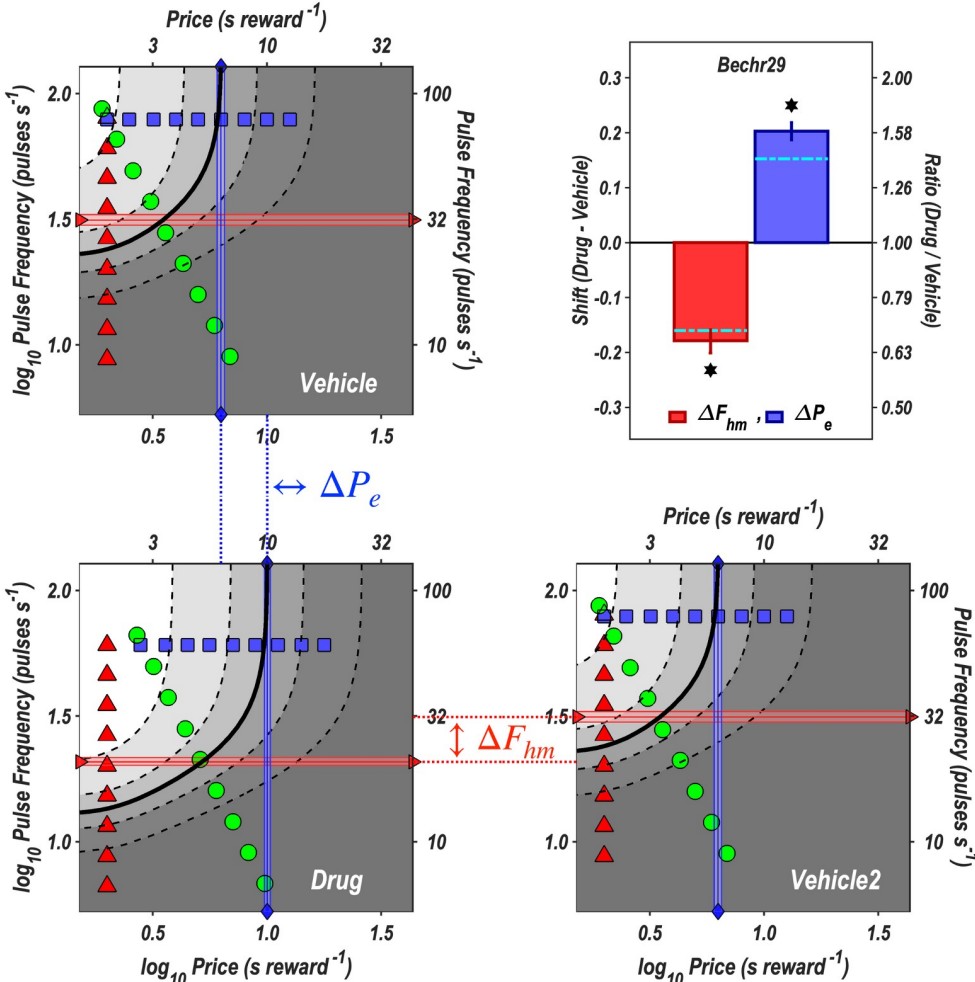

**Fig 7. Contour graphs of the reward-mountain surfaces fit to the vehicle and drug data from rat Bechr29.** The values of the independent variables along frequency sweeps are designated by red triangles, along price sweeps by blue squares, and along radial sweeps by green circles. The values of the location parameters, $P_{obj_e}$ and $F_{pulse_{hm}}$, are indicated by blue vertical lines with diamond end points and red horizontal lines with right-facing triangular end points, respectively. The shaded regions surrounding the lines denote 95% confidence intervals. The vehicle data are shown twice, once in the upper-left quadrant and once in the lower right. The dotted lines connecting the panels designate the shifts in the common-logarithmic values of the location parameters of the mountain, which are designated as $\{\Delta P_{obj_e}, \Delta F_{hm}\}$ and plotted in the bar graph in the upper-right panel. The dot-dash cyan lines superimposed on the bars show location-parameter estimates corrected for changes in frequency-following fidelity due to the displacement of the mountain along the pulse-frequency axis. (See section *Correction of the location-parameter estimates for changes in frequency-following fidelity* in the S1 File.) The 95% confidence intervals are shown in the bar graphs as vertical lines.

**$P_e$**: The estimates that locate the mountain along the price axis have also been corrected so as to remove the contribution of the changes in frequency-following fidelity due to the drug-induced displacement of the mountain along the pulse-frequency axis. (In the S1 File, please see sections Correction of the location-parameter estimates for changes in frequency-following fidelity, and Illustration of the correction for imperfect fre-quency-following fidelity). Table 3 gives the common logarithms of the corrected values ($P_{obj_e}^*$) listed in S9 Table along with the differences between the estimates for the drug and vehicle conditions and the 95% confidence intervals surrounding these differences. In all cases, the confidence band excludes zero, thus meeting our criterion for a statistically reliable effect. The signs of the differences are all

**Table 2. Drug-induced shifts of the reward mountain along the pulse-frequency axis.** The "$\log(F^*_{pulse_{hm}})$ Drg" and "$\log(F^*_{pulse_{hm}})$ Veh" columns list the common logarithms of the values in the "$F^*_{pulse_{hm}}$ Drg" and "$F^*_{pulse_{hm}}$ Veh" columns of S7 Table. The "diff" column shows the differences between the estimates for the drug and vehicle conditions. The "diff Lo" and "diff Hi" columns designate the lower and upper bounds of the 95% confidence interval surrounding these differences. The "⋆" character in the "excl 0" column indicates that zero falls outside the 95% confidence interval surrounding the estimates in the "diff" column. Differences so designated meet our criterion for statistical reliability. The rightmost column lists the value of $K_{da_{drug}}$ in Eqs S13,S14,S44 implied by the values in the "diff" column.

| Rat | $\log(F^*_{pulse_{hm}})$ Drg | $\log(F^*_{pulse_{hm}})$ Veh | diff | diff Lo | diff Hi | excl 0 | $K_{da_{drug}}$ |
|---|---|---|---|---|---|---|---|
| Bechr14 | 1.239 | 1.364 | -0.126 | -0.161 | -0.100 | ⋆ | 1.335 |
| Bechr19 | 1.015 | 1.160 | -0.145 | -0.206 | -0.088 | ⋆ | 1.395 |
| Bechr21 | 0.959 | 1.468 | -0.508 | -0.583 | -0.449 | ⋆ | 3.224 |
| Bechr26 | 1.195 | 1.339 | -0.144 | -0.165 | -0.121 | ⋆ | 1.393 |
| Bechr27 | 1.307 | 1.221 | 0.086 | 0.048 | 0.124 | ⋆ | 0.820 |
| Bechr28 | 1.119 | 1.431 | -0.313 | -0.331 | -0.295 | ⋆ | 2.055 |
| Bechr29 | 1.260 | 1.420 | -0.160 | -0.179 | -0.142 | ⋆ | 1.446 |

positive, indicating that under the influence of dopamine/transporter blockade, a higher price was required to bring time allocated to pursuit of a maximal reward to its middle value. With one exception (Rat Bechr27), the correction reduced the magnitude of the drug/induced shift (as shown by the position of the dot-dash cyan lines in the bar-graph panels of Fig 7 and S23-S28 Figs).

The corrected values of the two location parameters are shown in Fig 8.

The corrected values of the location parameters are uncorrelated in both the vehicle ($\rho = 0.37$, $p = 0.41$) and drug ($\rho = -0.24$, $p = 0.60$) conditions, as are the drug-induced **shifts** in these values ($\rho = -0.2830$; $p = 0.5385$, S29 Fig).

All parameters of the best-fitting models for each rat are shown in S10-S15 Tables in the S1 File. (The values of the location parameters in these tables are uncorrected).

## Histology

Immunohistology and confocal microscopy (Fig 9) confirms that the tips of all optical-fiber implants (heavy, angled black lines) were located close to transfected midbrain-dopamine neurons and that ChR2 was selectively expressed in TH-positive neurons.

**Table 3. The location of the reward mountain along the price axis.** The "$\log(P^*_e)$ Drg" and "$\log(P^*_e)$ Veh" columns list the estimated values that the $\log_{10}\left((P_{obj_e})\right)$ parameter would have attained in the drug and vehicle conditions, respectively, had frequency following been perfect. The "diff" column shows the differences between the estimates for the drug and vehicle conditions, whereas the "diff Lo" and "diff Hi" columns designate the lower and upper bounds of the 95% confidence interval surrounding these differences. The ⋆ character in the "excl 0" column indicates that zero falls outside the 95% confidence interval surrounding the estimates in the "diff" column. Differences so designated meet our criterion for statistical reliability. The rightmost column lists the proportional drug-induced change in the value of $P^*_e$ corresponding to the values in the "diff" column. ($\Delta P^*_e = 10^{diff}$).

| Rat | $\log(P^*_e)$ Drg | $\log(P^*_e)$ Veh | diff | diff Lo | diff Hi | excl 0 | $\Delta P^*_e$ |
|---|---|---|---|---|---|---|---|
| Bechr14 | 0.981 | 0.787 | 0.194 | 0.158 | 0.234 | ⋆ | 1.506 |
| Bechr19 | 1.070 | 0.730 | 0.340 | 0.317 | 0.365 | ⋆ | 2.174 |
| Bechr21 | 1.400 | 1.136 | 0.264 | 0.243 | 0.289 | ⋆ | 1.576 |
| Bechr26 | 1.148 | 1.065 | 0.082 | 0.063 | 0.101 | ⋆ | 1.211 |
| Bechr27 | 1.351 | 1.289 | 0.061 | 0.049 | 0.074 | ⋆ | 1.177 |
| Bechr28 | 1.877 | 1.709 | 0.168 | 0.139 | 0.195 | ⋆ | 1.354 |
| Bechr29 | 1.002 | 0.799 | 0.203 | 0.187 | 0.218 | ⋆ | 1.421 |

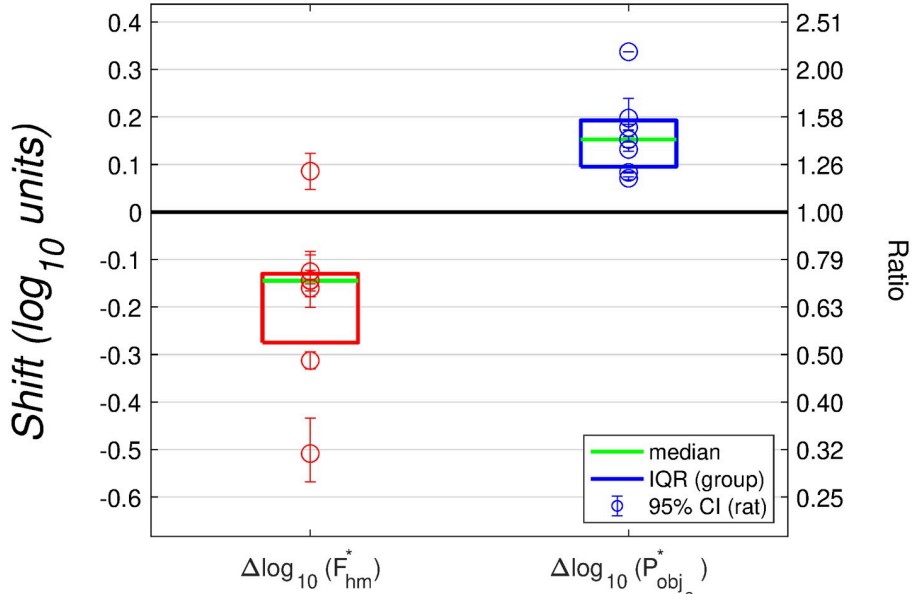

**Fig 8. Drug-induced shifts in the location parameters of the reward mountain.** The $\Delta log_{10}(F_{hm}^*)$ values give the shifts of the reward-growth function and reward mountain along the pulse-frequency axis, whereas the $\Delta log_{10}(P_{obj_e}^*)$ values give the shifts along the price axis. These values have been corrected for the estimated change in frequency-following fidelity due to the drug-induced displacement of the reward mountain along the pulse-frequency axis. According to the reward-mountain model, the $\Delta log_{10}(F_{hm}^*)$ values reflect action of the dopamine transporter blocker prior to the input to the reward-growth function, whereas the $\Delta log_{10}(P_e^*)$ values reflect drug action at, or beyond, the output of the reward-growth function. $\Delta log_{10}(F_{hm}^*)$ is shorthand for $\Delta log_{10}(F_{pulse_{hm}}^*)$.

## Discussion

Decades of psychophysical research on eICSS have provided a detailed portrait of the neuro-computational processes that translate a train of electrical current pulses into subsequent reward-seeking behavior. The reward-mountain model [37–39] integrates these findings so as to predict the allocation of behavior to eICSS from the strength and cost of the rewarding stimulation. The Matlab® live script documented in the S1 File simulates sev-eral of the principal validations studies, and compares these simulations to empirical results. The simulations and empirical data show that rescaling the input to the reward-growth func-tion (tantamount to changing its ***input*** sensitivity (Fig 1)) shifts the reward mountain along the pulse-frequency axis, whereas rescaling performed at, or beyond, the output of the reward-growth function (tantamount to changing its ***output*** amplification (Fig 1)) shifts the reward mountain along the price axis.

Substitution of optical for electrical stimulation [50–52] offers significant advantages: the resulting neural activation is confined to a known, genetically specified, neural population, and the neurons that express the light-sensitive transducer molecule are readily visualized. However, few psychophysical data have been reported concerning the processes that translate the optical input into observable reward-seeking behavior. The results of the present study demonstrate that the dependence of oICSS and eICSS on the strength and cost of rewarding stimulation is formally similar. Nonetheless, boosting dopaminergic neurotransmission by means of dopamine/transporter blockade alters oICSS of midbrain dopamine neurons and eICSS of the MFB in strikingly different ways. We show here that the reward mountain is shifted along the pulse-frequency axis by the specific dopamine-transporter blocker, GBR-

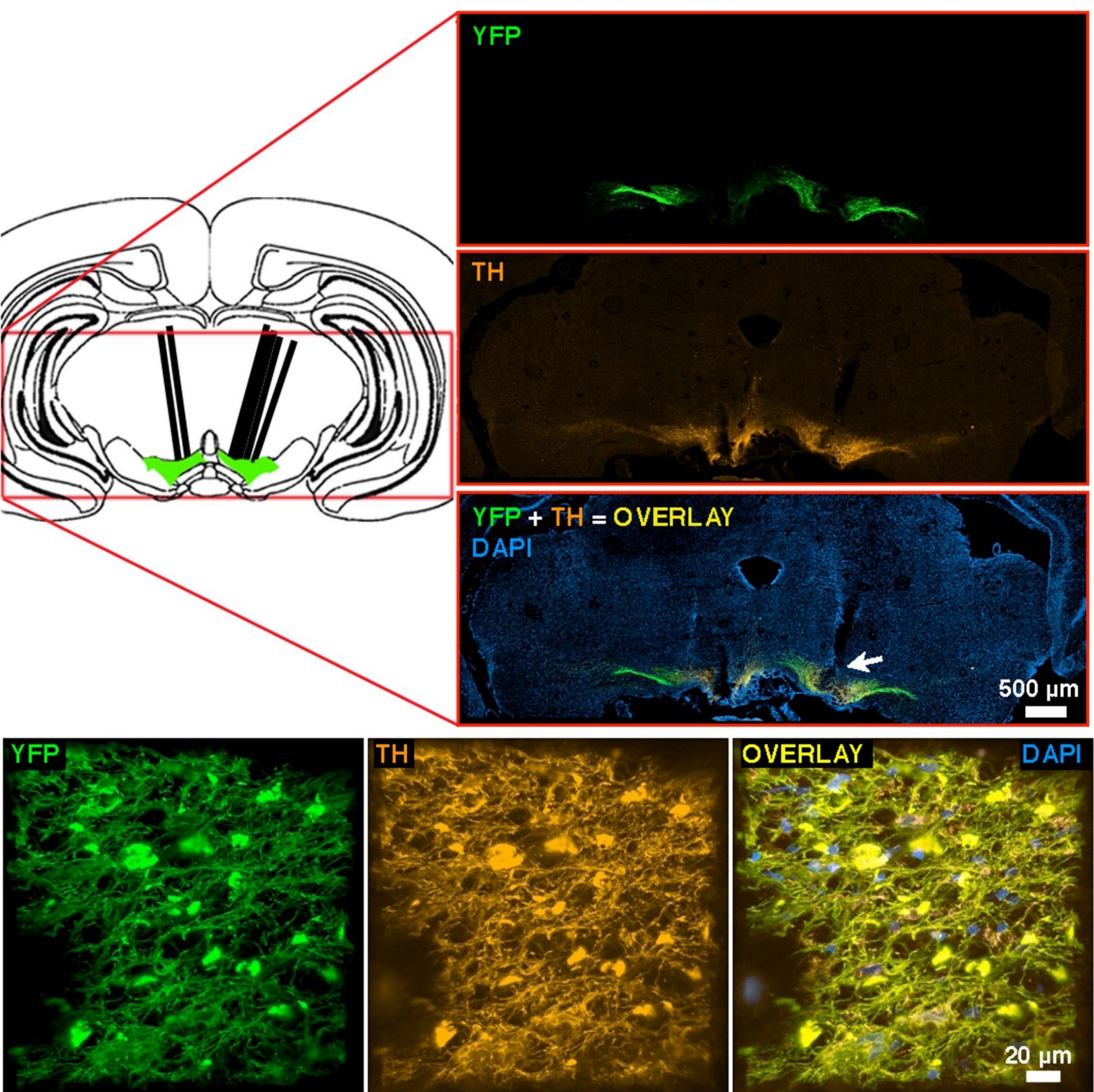

**Fig 9. Images of viral construct expression from rat BeChr29. upper left**: Schematic representation of a coronal section at the approximate location of the optical-implant tip (modified from [56]). Heavy angled black lines denote the optical-fiber tracks. **upper right**: Representative immunohistochemical images. YFP and TH staining is shown in the top and middle panels respectively. The bottom panel shows the co-expression of YFP and TH (overlay) along with DAPI for anatomical reference. The arrow indicates the approximate location of the tip of the optical implant used for oICSS. **bottom** High magnification 3D reconstruction of the area below the optical-fiber track. Left: YFP-positive neurons; Middle: TH-positive neurons; Right: Overlay of YFP, TH and DAPI staining. Note that YFP is expressed in TH-positive neurons.

12909. Thus, the transporter blocker acted as if to rescale the input to the reward-growth function for oICSS (i.e., via an action at, or prior to, the input). According to the series-circuit model, the directly activated neurons subserving eICSS of the medial forebrain bundle produce their rewarding effect by transsynaptic activation of the same midbrain dopamine neurons that were activated directly in the present study by optogenetic means. If so, specific dopamine transporter blockade must also shift the reward-growth function for eICSS along the pulse-frequency axis. However, it failed to do so in all 10 subjects of the study in which eICSS was challenged by the administration of GBR-12909 [34].

The reward-mountain model was developed to account for data from eICSS experiments. We begin by considering what the current results tell us about generalization of this model to oICSS. We then discuss the significance of the location-parameter values that position the surfaces fitted to the vehicle data within the space defined by the strength and cost of the optical reward. Next, we turn to the main experimental question posed in this study: how does blockade of the dopamine transporter alter the position of the reward mountain? Finally, we discuss the implications of those drug-induced shifts, and we thereby show why a new way of thinking about brain reward circuitry is required to integrate the results of the pharmacological challenge with existing eICSS data. This new perspective entails the parallel flow of reward signals in dopaminergic and non-dopaminergic pathways prior to their convergence on the behavioral final common path.

### The reward-mountain model generalizes successfully to oICSS

The shape of the reward-mountain surface fitted to the oICSS data (Fig 6, S17-S22 Figs) resembles that of the surfaces fit to eICSS data reported previously [8, 34, 38–40, 46, 47, 57]. Time allocation falls as pulse-frequency is decreased and/or price is increased. The resemblance between the shapes of the surfaces fitted to prior eICSS and current oICSS data suggests that the reward-mountain model generalized well to the case of oICSS and that pursuit of rewarding optical or electrical stimulation depends similarly on the strength and cost of reward.

Embedded within the reward-mountain model derived in the context of eICSS studies is the notion that reward intensity depends on the aggregate rate of firing induced in the directly stimulated neurons by a pulse train of fixed duration [20–23, 33]. A definitive test of this "counter model" [19] has yet to be reported in the case of oICSS. Nonetheless, there are indications that the counter model holds in this case as well.

Ilango and colleagues trained mice to press a lever to receive optical stimulation of ChR2-expressing midbrain dopamine neurons [58]. They showed that lever-pressing rates increased as a function of both pulse duration and optical power, two variables that conjointly determine the number of neurons activated by the optical stimulation. Lever-pressing also increased when an 8-pulse train was delivered at progressively higher pulse frequencies. This latter result cannot be linked unambiguously to the counter model because the train duration covaried with the pulse frequency; in the case of eICSS, the formal relationships between these two variables and reward intensity differ [33]. The frequency-sweep data reported here were obtained with train duration held constant. Thus, they complement and extend the findings of Ilango and colleagues while avoiding the complication of covariation between the pulse frequency and train duration. Time allocation increased systematically as a function of pulse frequency, as the reward-mountain model and the counter model embedded within it predict.

Ilango and colleagues also measured changes in lever-pressing rates during performance of oICSS on fixed-interval or fixed-ratio schedules of reinforcement [58]. Response rates declined as either the minimum inter-reward interval or the required number of responses per reward was increased.

Fixed-interval schedules set the minimum inter-reward interval, but the subject determines the physical effort expended in harvesting the reward. Although only a single response is required to trigger reward delivery after the fixed interval has elapsed, subjects typically begin responding much earlier and thus expend more effort than is strictly necessary [59]. Fixed-ratio schedules set the number of responses required to trigger reward delivery but cede control of the minimum inter-reward interval to the subject, who can vary response rate so as to bring reward delivery closer (or further) in time, as Ilango and colleagues observed. In contrast to these two classic schedules, the cumulative handling-time schedule employed here fixes both the minimum inter-reward interval and the rate of physical exertion required to secure a reward. This makes it possible to vary the opportunity cost of the reward independently of the required rate of physical exertion, as was done here and in previous experiments carried out in the reward-mountain paradigm. As in the case of the eICSS experiments, increases in the opportunity cost (price) of the reward decreased time allocation systematically (Fig 5, S7-S12 Figs, Fig 7, S23-S28 Figs).

## Vehicle condition: Location parameters and their significance

**Pulse-frequency axis**: In rats working for fixed-duration trains of rewarding, electrical, MFB stimulation, the parameter that positions the reward mountain along the pulse-frequency axis, $F_{pulse_{hm}}$, depends on the stimulation current [37]. This variable, (in conjunction with the pulse duration [18, 60]) determines the number of directly stimulated neurons [19] in a manner dependent upon their excitability and spatial distribution with respect to the electrode tip. By analogy, we expect the number of midbrain dopamine neurons recruited by optical pulses and trains of fixed duration to depend on the optical power, the level of ChR2 expression in the dopamine neurons, and the spatial distribution of these neurons with respect to the tip of the optical probe. Given the inevitable variation in ChR2 expression and probe-tip location, it is not surprising that that $F_{pulse_{hm}}$ varied over a wide range in the vehicle condition, from 16.3–35.8 pulses s$^{-1}$ (S7 Table). Even after correction for differences in frequency-following fidelity, the estimated firing frequencies induced by the $F_{pulse_{hm}}$ values (i.e., $F^*_{pulse_{hm}}$) vary more than twofold.

**Price axis**: In principle (and in contrast to $F^*_{pulse_{hm}}$), the corrected estimate of the parameter that positions the eICSS reward mountain along the price axis, $P^*_{obj_e}$, can be compared meaningfully across subjects and electrical stimulation sites. The validity of this comparison rests on the assumption that both the value of alternate activities, such as grooming, resting, and exploring, and the effort cost of performing the lever-depression task do not vary substantially across subjects or stimulation sites. If so, then the price at which a maximally intense, stimulation-generated reward equals the value of alternate activities ($P^*_{obj_e}$) reflects the maximum intensity of the rewarding effect, independent of the value of the stimulation parameters required to drive reward intensity to its upper asymptote. On this view, the rat is willing to sacrifice more leisure in order to obtain strong stimulation of a "good" eICSS site than a poorer one.

Could such a comparison extend meaningfully to different forms and targets of stimulation? The current oICSS experiment was carried out in the same testing chambers as previous eICSS studies. If the value of alternate activities in that environment does not change systematically as a function of whether electrical stimulation of the MFB or optical stimulation of midbrain dopamine neurons serves as the reward, then it is of interest to compare the $P^*_{obj_e}$ values of the current study to the $P_{obj_e}$ values obtained in prior eICSS experiments. (For reasons discussed in section Correction of the location-parameter estimates for changes in frequency-

following fidelity of the S1 File, the $P_{obj_e}$ estimates from the eICSS studies do not typically require correction for imperfect frequency-following fidelity). The results of this comparison are intriguing.

The median of the corrected $P^*_{obj_e}$ values for the vehicle condition of the present study, 11.8 s, is marginally greater than the median of the 54 values reported in previous eICSS studies [8, 34, 38–40, 46, 47, 57], but the range is much more extreme. Whereas $P_{obj_e}$ values typically vary over a roughly twofold range in the eICSS studies, the $P^*_{obj_e}$ values vary over a more than a tenfold range in the current oICSS dataset. The maximum (56.4) is much greater than any value obtained in the eICSS studies, whereas the minimum (5.4) is smaller.

According to the reward-mountain model, the large magnitude of the maximum $P^*_{obj_e}$ value means that direct optical stimulation of midbrain dopamine neurons can produce a much more potent rewarding effect than electrical stimulation of the MFB, at least in an extreme case. Perhaps the maximal intensity of the rewarding effects reported here varies so profoundly across subjects because the activation of different subsets of dopamine neurons is rewarding for different reasons. If so, the large range of $P^*_{obj_e}$ values would add to the accumulating evidence for functional heterogeneity of midbrain dopamine neurons [61–68].

The reward mountain was measured in only seven subjects in the current study, and the dispersion of the tips of the optical probes is modest. Given these limitations and the uncertainly inherent in estimating the location of the optically activated dopamine neurons from the position of the tip, we cannot provide a meaningful assessment of whether the maximum intensity of the rewarding effect, as indexed by the $P^*_{obj_e}$ value, is correlated with the location of the optically activated dopamine neurons. It would be interesting to pursue this issue in a larger number of subjects and across a larger region of the midbrain. Mice can be trained to perform oICSS for stimulation of sites arrayed across much of the medio-lateral extent of dopaminergic cell bodies in the ventral midbrain, from the lateral portion of the substantia nigra to the medial boder of the ventral tegmental area [69]. Does $P^*_{obj_e}$ vary systematically as a function of tip location across the full extents of the midbrain region where dopaminergic cell bodies are located?

The reader may be tempted to ascribe the large variation in $P^*_{obj_e}$ values to variables such as across-subject differences in ChR2 expression and the location of the optical-fiber tip. If, as we argue below, the form of the reward-growth function resembles a logistic, such differences would have contributed to the variation in $F^*_{pulse_{hm}}$ values and not to the $P^*_{obj_e}$ values. That is so because across-subject differences in ChR2 expression and tip location influence the **_input_** to the reward-growth function: the aggregate firing rate induced by the optical stimulation in the midbrain dopamine neurons. In contrast, the scalar that determines the maximum reward intensity ($K_{rg}$) alters the **_output_** of the reward-growth function (Fig 3), and is one of the components of the $P^*_{obj_e}$ parameter (Eq S30). Thus the reward-mountain model holds that the value of $P^*_{obj_e}$ reflects the maximum intensity of the rewarding effect.

According to the argument that the $F^*_{pulse_{hm}}$ values, but not $P^*_{obj_e}$ values, should depend on local conditions such as ChR2 expression and the spatial distribution of dopamine neurons with respect to the probe tip, the values of these two location parameters should be uncorrelated. The results are consistent with this expectation.

## Drug-induced shifts in the position of the reward mountain

Table 2, S8, S9 Tables show that the dopamine/transporter blocker, GBR-12909, shifted the reward mountain reliably along the pulse-frequency and price axes in all 7 rats. (Only one

result is discrepant with those from the rest of the group: the direction of the shift in the data from rat Bechr27, which is also the smallest of the shifts along the pulse-frequency axis.) The magnitudes of the shifts along the two axes are uncorrelated. These results have multiple implications:

1. The reward-growth function responsible for oICSS has independent input-scaling and output-scaling parameters and resembles a logistic.

2. The shifts along the pulse-frequency axis are due to an effect of the drug at, or prior to, the input to the reward-growth function.

3. The shifts along the price axis are due to an independent effect of the drug at, or beyond, the output of the reward-growth function.

4. The shifts along the pulse-frequency axis create grave problems for the hypothesis that the rewarding effect produced by electrical stimulation of the MFB arises solely from transsynaptic activation of midbrain dopamine neurons (the series-circuit model).

**The reward-growth function for oICSS.** The effects of dopamine-transporter blockade on the reward mountain powerfully constrain the underlying reward-growth function, and the functional form implied by these results has far-reaching implications concerning the structure of brain-reward circuitry. We develop this argument by first analyzing the form of the reward-growth function underlying the reward-mountain model, then by demonstrating how the data support this form, and finally by deriving the implications of this functional form for the structure of brain-reward circuity.

The cross-sectional shape of the reward-mountain is determined principally by the form of the reward-growth function. If frequency-following fidelity were perfect and subjective prices were equal to objective ones, then the contour lines would have the same form as the reward-growth function (flipped and rotated, as shown in Fig 10 of [40].) This is why orthogonal shifts in position of the reward-growth function (Fig 1) are expressed in correspondingly orthogonal shifts in the position of the reward mountain.

The subjective-price function [8] bends the contour lines towards the horizontal at very low prices but has no effect on the location of the reward mountain within the coordinates defined by the independent variables (price and pulse frequency). As estimated by Solomon et al. [40], this function converges on the objective price, and the discrepancy between the subjective and objective values is within 1% once the objective price exceeds 3.18 s. The subjective and objective prices can no longer be distinguished once the objective price approaches the values of the $P^*_{obj_e}$ location parameter (S9 Table).

Given the frequency-following function assumed here, frequency-following fidelity is imperfect over much or all of the tested range of pulse frequencies. That said, firing frequency falls only slightly short of the pulse frequency at pulse frequencies below the $F_{pulse_{hm}}$ values (S13 Fig). The procedure for generating the corrected values of the parameter that locates the reward mountain along the pulse-frequency axis ($F^*_{pulse_{hm}}$) is designed to remove the influence of imperfect frequency-following fidelity. We emphasize that the correction procedure adjusts the estimates of the shifts but cannot manufacture these out of whole cloth. The correction is driven by differences in the location along the pulse-frequency axis of the surfaces fit to the vehicle and drug data. Had the drug failed to displace the mountain surface, the correction would be zero. Although the correction is likely imperfect, the estimated drug-induced displacement of the reward mountain along the pulse-frequency axis (Table 2) should be due largely to the effect of the drug on the reward-growth function.

Unlike the case for eICSS [21–23, 28], the reward-growth function for oICSS has not yet been measured directly. The fact that a reward-mountain surface based on a logistic reward-growth function fits the current data well provides one hint that this function may indeed be logistic in form, or very similar. But there is a deeper sense in which the results of the present study provide crucial new information about the reward-growth function for oICSS: The displacement of the reward mountain along both the strength and cost axes by dopamine/transporter blockade **requires** that the reward-growth function for oICSS have independent location-scaling and output-scaling parameters, like the logistic reward-growth function for eICSS. To explain why this is so, it is helpful to reformat the logistic reward-growth equation (Eq S11), as follows:

$$\frac{R_{bsr}}{K_{rg}} = \frac{\left(\frac{F_{firing}}{F_{firing_{hm}}}\right)^g}{\left(\frac{F_{firing}}{F_{firing_{hm}}}\right)^g + 1}$$

where

$F_{firing}$ = firing rate produced by a pulse frequency of $F_{pulse}$ pulse $s^{-1}$ (1)

$F_{firing_{hm}}$ = firing rate required to drive reward intensity to half its maximum value

$g$ = exponent that determines the steepness of reward − intensity growth as a function of pulse frequency

$K_{rg}$ = reward − growth scalar

$R_{bsr}$ = reward intensity produced by $F_{firing}$

This format makes clear that $K_{rg}$ scales the **output** of the logistic reward-growth function ($R_{bsr}$), whereas $F_{firing_{hm}}$ scales the **input** ($F_{firing}$). These two scalars act independently, as Fig 10 illustrates: changes in $F_{firing_{hm}}$ shift the reward-growth function in the log-log plot in the upper-left panel along the X (pulse-frequency) axis, whereas changes in $K_{rg}$ shift the log-log plot of the reward-growth function in the upper-right panel along the Y (reward-intensity) axis. The inserts in the upper panels of Fig 10 and the contour and bar graphs in Fig 7, S30, S31 Figs show that these orthogonal shifts move the reward mountain along the pulse-frequency and price axes, respectively.

What would happen if frequency-following fidelity remained the same, but the input-scaling and output-scaling parameters of the reward-growth function were no longer independent? We can simulate such a case by replacing the logistic reward-growth function in Eq 1 with a power function:

$$\frac{R_{bsr}}{K_{out}} = \left(\frac{F_{firing}}{K_{in}}\right)^g$$

where

$K_{in}$ = the input − scaling parameter (2)

$K_{out}$ = the output − scaling parameter

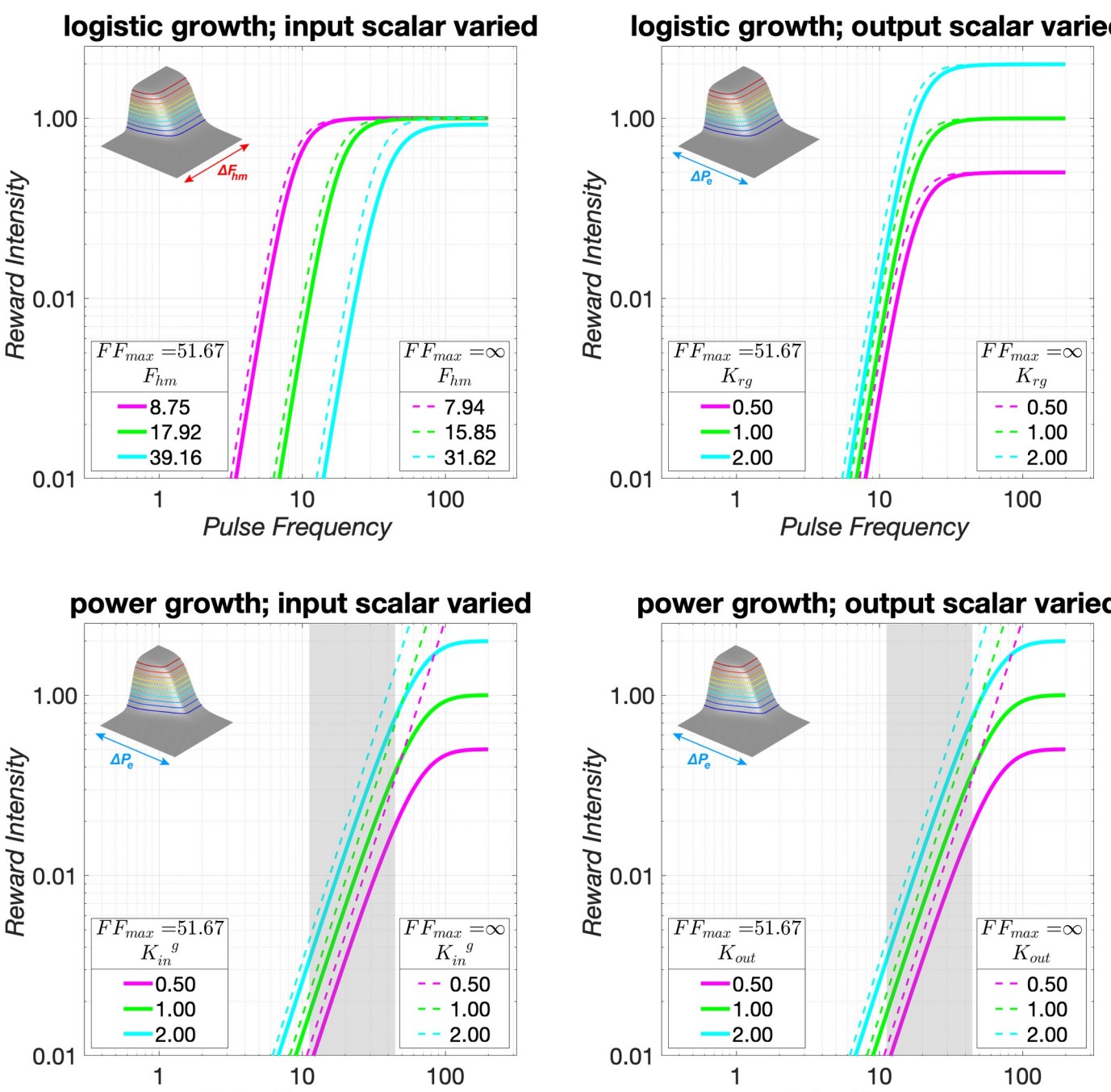

**Fig 10. Influence of input- and output-scaling parameters on reward-growth functions.** The upper panels show logistic reward-growth functions, whereas the lower panels show power functions. Thus, the upper panels are analogous to Fig 1, but in double-logarithmic, rather than semi-logarithmic, coordinates, and using pulse-frequency values typical of oICSS rather than eICSS experiments. Solid lines map the ***pulse*** frequency into the reward intensity using the assumed frequency-following function (see: Parameters of the frequency-following function for oICSS). $FF_{max}$ is the maximum firing frequency ($F_{firing_{max}}$) attainable given the form and parameters of the assumed frequency-following function. The dashed lines map the ***firing*** frequency into the reward intensity. (These lines are drawn assuming perfect frequency-following fidelity ($FF_{max} = \infty$), and thus pulse frequency and firing frequency are equivalent.) In the case of the logistic reward-growth functions in the upper panels, changing the values of the input- and output-scaling parameters produces independent effects, shifting the reward-growth function in orthogonal directions. In contrast, changing the values of the input- and output-scaling parameters produces identical effects on the power-growth functions plotted in the lower panels.

This power-growth function can be rewritten as

$$\frac{R_{bsr}}{K_{out} \times K_{in}{}^{-g}} = F_{firing}{}^{g} \tag{3}$$

In contrast to the case of logistic growth, the two scaling constants act jointly, in an inseparable manner, and exclusively on the output of the function. Changing either scaling constant shifts the power-growth function along the Y axis but does not change its position along the X axis (lower panels of Fig 10). In contrast to the logistic function (upper panels), the two constants act as one. S33 Fig shows that changing the value of the input-scaling parameter of the power-growth function moves the reward mountain only along the price axis, and S34 Fig shows that changing the value of the output-scaling parameter has the same effect.

The three power-growth functions in the lower panels of Fig 10 designated by solid lines all bend at the same location along the X axis. This is because the frequency-following function that translates pulse-frequency into firing frequency (Eq S1, S13 Fig) levels off after that point, truncating the input. However, despite this common landmark, these functions lack a true location parameter: as shown by the dashed lines, the underlying reward-growth function, which maps the firing frequency into reward intensity, rises continuously over the entire domain. In contrast, the logistic curves in the upper-left panel of Fig 10 level off at different locations along the X axis, as determined by their $F_{firing_{hm}}$ (and $g$) values.

A location parameter analogous to $F_{firing_{hm}}$ can be defined for any monotonic growth function that has a saturation point, like the logistic reward-growth function for eICSS described by the matching data obtained by Gallistel's group [21–23, 28]. Such curves have an implicit threshold pulse frequency that just suffices to drive reward intensity out of the baseline noise and a saturating pulse frequency beyond which further increases cannot drive reward intensity higher. The growth of reward intensity is constrained to occur over this interval. The width of the interval between the threshold and saturating pulse frequencies is determined by the reward-growth exponent, $g$ (Eq S11), whereas the location of this growth region along the function's domain is set by the input-scaling parameter, $F_{firing_{hm}}$. In this study, dopamine/transporter blockade shifted the reward mountain reliably along the pulse-frequency axis (Table 2, Fig 7, S23-S28 Figs). A reward-growth function that lacks independent input- and output-scaling parameters cannot produce such results, as the lower panels of Fig 10) and S33 Figs, S34 Fig illustrate. Instead, a logistic-like function is required.

**Shifts along the pulse-frequency axis.** Blockade of the dopamine transporter by GBR-12909 increases the amplitude of stimulation-induced dopamine transients recorded in the nucleus-accumbens terminal field [70, 71]. It follows that in the current experiment, a lower pulse frequency will suffice to drive peak dopamine concentration to a given level under the influence of the drug than in the vehicle condition. Transporter blockade would thus rescale upwards the input to the reward-growth function. Such an effect is illustrated in S16 Fig. The amplification of the stimulation-induced dopamine release by GBR-12909 is represented by the triangle labelled "$K_{da}$." The drug rescales upwards the impact of the phasic stimulation-induced increase in the aggregate firing rate ("DA drive") on the input to the logistic-like reward-growth function, which is thus shifted leftwards along the pulse-frequency axis (Fig 10), dragging the reward-mountain surface downward along this axis (Fig 7). New optical methods [72, 73] facilitate concurrent measurement of dopamine concentrations and behavioral allocation to reward pursuit and could thus provide a direct test of the proposed mechanism for the observed decreases in $F^{*}_{pulse_{hm}}$.

**Shifts along the price axis.** In addition to increasing the amplitude of stimulation-induced dopamine transients, blockade of the dopamine transporter by GBR-12909 also

increases the baseline ("tonic") level of dopamine [34, 70, 71]. The increase in dopamine tone could rescale the output of the reward-growth function upwards, reduce subjective effort costs and/or diminish the value of activities that compete with pursuit of the optical reward. Such effects could arise from increases in $K_{rg}$ and/or decreases in $K_{ec}$ or $K_{aa}$ (S16 Fig, Eqs S10, S11, S19,). Any combination of these effects could account for the observed rightward shifts of the reward mountain along the price axis (Fig 7, S23-S28 Figs). This can be seen by reformatting Eq S30 and expressing it verbally as follows:

$$P_{sub_e} = \frac{\text{maximum reward intensity}}{\text{subjective effort cost} \times \text{value of competing activities}} \tag{4}$$

(Reward probability has been omitted from Eq 4 because the probability of reward upon satisfaction of the response requirement was equal to one in the present study).

Teasing apart the three alternative accounts of the drug-induced change in $P^*_{obj_e}$ may not be feasible on the basis of behavioral data alone; it will likely require identification of the neural substrates underlying each of these influences and measurement of how dopamine-transporter blockade influences signal flow within each of these circuits.

**Implications for the series-circuit model of brain-reward circuitry.** According to the series-circuit model (Fig 3), the rewarding effects produced either by electrical stimulation of the MFB or by optical stimulation of midbrain dopamine neurons arise from a common cause: activation of the dopamine neurons. The optical stimulation excites these neurons directly, whereas the electrical stimulation activates them transsynaptically. We have explained above that a reward-growth function with independent input-scaling and output-scaling parameters lies downstream from the dopamine neurons responsible for oICSS. If so, boosting the peak stimulation-induced dopamine concentration by means of transporter blockade should shift the contour map of the reward mountain downwards along the pulse-frequency axis in both prior eICSS experiments and the current oICSS experiment. It does not. Whereas the predicted shift is seen in six of seven cases in the current oICSS results, reliable shifts in this direction are absent in all eight cases reported in the analogous eICSS study [34]. Similarly, reliable, downwards shifts were absent in seven of eight eICSS subjects tested under the influence of AM-251, a cannabinoid CB-1 antagonist that attenuates stimulation-induced dopamine release [46] and all six eICSS subjects tested under the influence of the dopamine-receptor blocker, pimozide [47]. This sharp discrepancy is highly problematic for the series-circuit model. If dopamine release in the terminal fields of midbrain dopamine neurons is the sole and common cause of the rewarding effect produced by electrical stimulation of the MFB and optical stimulation of midbrain dopamine neurons, why does the reward mountain respond so differently to perturbation of dopamine neurotransmission in the eICSS and oICSS studies?

Could the different mechanisms by which dopamine neurons are activated explain why dopamine-transporter blockade fails to shift the reward mountain along the pulse-frequency axis in rats working for electrical stimulation of the MFB but succeeds in doing so in rats working for optical stimulation of the ventral midbrain? We doubt that this is a viable explanation. To explain why we must first lay out in some detail this argument for rescuing the series-circuit hypothesis.

Given that ChR2 is found throughout the excitable regions of the cell membrane and that the optical probe is ~10x larger than the soma of a dopamine neuron, the optically generated spikes likely arise downstream from the somatodendritic region. If dopamine-transporter blockade increased extracellular dopamine concentrations in the somatodendritic region, this would increase binding of dopamine to D2 autoreceptors and potentially decrease the

sensitivity of these neurons to synaptic input. Such an effect could well be bypassed in the case of oICSS due to the activation of the ChR2-expressing neurons downstream from the somato-dendritic region.

In order for such an explanation to account for the failure of GBR-12909 to shift the reward mountain along the pulse-frequency axis in rats working for electrical stimulation of the MFB, the putative autoreceptor-mediated inhibition would have to exactly counteract the drug-induced enhancement of dopaminergic neurotransmission in the terminal fields of those dopamine neurons that received sufficient excitatory synaptic drive to overcome the influence of the D2 stimulation. (Recall that the reward mountain was not shifted reliably along the pulse-frequency axis in either direction in all eight subjects in the eICSS study.) Such exact counterbalancing seems unlikely.

The impact of the dopamine transporter is very different in the somatodendritic and terminal regions of dopamine neurons. The rate of dopamine reuptake is as much as 200 times higher [74] and dopamine-transporter expression from 3–10 times greater [74, 75] in the terminal region than in the somatodendritic region. In a study carried out in guinea-pig brain slices, GBR-12909 failed to increase electrically induced release of dopamine in the VTA cell-body region [76], in contrast to its potent augmentation of release in terminal regions. This drug greatly boosted the amplitude of dopamine transients recorded voltammetrically in the nucleus accumbens terminal field of rats working for rewarding electrical stimulation of the VTA [70]. A recent study [77] shows very similar release of dopamine in the nucleus accumbens of rats working for electrical stimulation of either the VTA or of the MFB site used in the eICSS study of the effect of GBR on the reward mountain [34], suggesting that GBR-12909 would boost release of dopamine in the nucleus accumbens in response to electrical stimulation of the MFB site employed in the study of the effect of dopamine-transporter blockade on the position of the reward mountain [34]. Moreover, cocaine, which also blocks the dopamine transporter, greatly increased release of dopamine in the nucleus-accumbens shell in response to transsynaptic activation of midbrain dopamine neurons by electrical stimulation of the laterodorsal tegmental area [78, 79]. Taken together, the evidence makes it highly unlikely that in the reward-mountain study by Hernandez et al. [34], autoreceptor-mediated inhibition prevented GBR-12909 from boosting phasic release of dopamine in the terminal fields of midbrain dopamine neurons. Thus, the challenge posed by the present data to the series-circuit model stands, and alternative accounts must be explored.

## Toward a new model of brain-reward circuitry

To account for both the eICSS and oICSS data in the simulations, we propose development and exploration of models in which the reward-intensity signals evoked by electrical stimulation of the MFB and by optical stimulation of midbrain dopamine converge on a final common path. In such models, distinct reward-growth functions translate aggregate firing rate in the MFB neurons and the dopamine neurons into reward intensity. One such model is shown in Fig 11. The two reward-intensity signals (the output of the two reward-growth functions) converge onto the final common path underlying the behaviors entailed in approaching and holding down the lever. Thus, we are proposing that reward-related signals flow in parallel through a portion of the brain reward circuitry subserving ICSS and that the reward-intensity signal in the MFB limb of the circuit bypasses the midbrain dopamine neurons en route to the final common path.

Convergence models face a seemingly daunting challenge: electrical stimulation of the MFB activates midbrain dopamine neurons [71, 80–83]. If so, one would expect such stimulation to drive signaling in both of the hypothesized converging pathways. Wouldn't this produce at

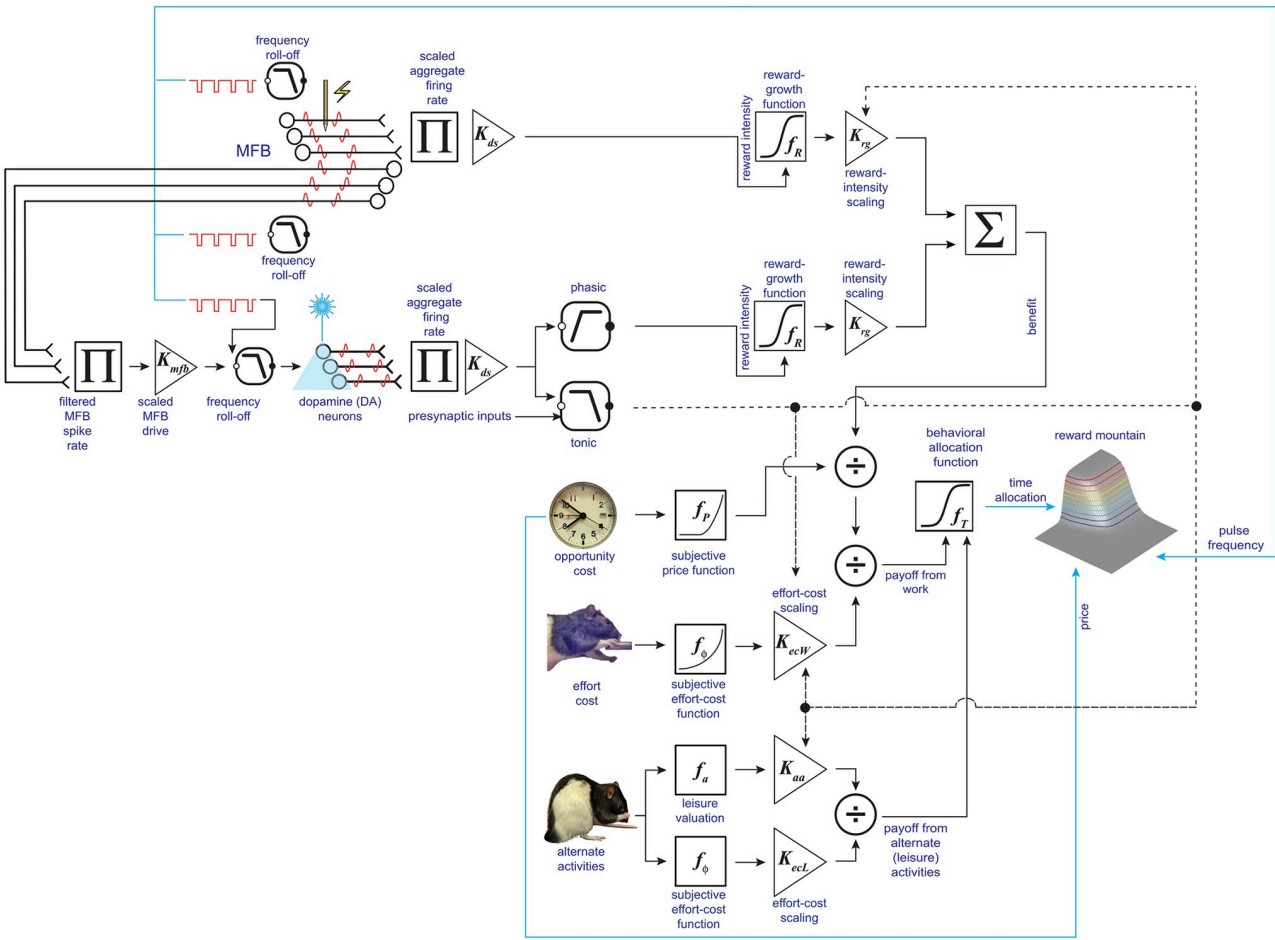

**Fig 11. Parallel pathways conveying reward-intensity signals to a final common path.** The directly activated neurons subserving eICSS of the MFB are depicted at the left of the top row of symbols. The electrode also activates a second population of fibers, with lower frequency-following fidelity [80], that project directly or indirectly to midbrain dopamine neurons, which have yet lower frequency-following fidelity. Optical stimulation of midbrain dopamine neurons activates only the lower limb of the circuit. The outputs of the reward-growth functions in the two limbs converge (Σ) on the final common path for reward evaluation and pursuit. The reward-growth functions are shown here as S-shaped, which is the form they describe when plotted in linear or semi-logarithmic (reward intensity versus *log*(pulse frequency)) coordinates.

least some displacement of the reward mountain along the pulse-frequency axis in response to dopamine-transporter blockade? The simulations documented in the accompanying Matlab® Live Script show that this is not necessarily the case. Indeed, the simulations show that given reasonable assumptions and values drawn from the current data, a convergence model can replicate the findings reported here. The following paragraphs explain how.

In the eICSS studies entailing measurement of the reward mountain under the influence of drug-induced changes in dopamine neurotransmission, the train duration was 0.5 s, whereas it was 1.0 s in the current oICSS study. Gallistel [18] and Sonnenschein and colleagues [33] showed that the pulse frequency required to sustain half-maximal eICSS performance is a rectangular hyperbolic function of train duration. As shown in the Live Script, we used a rectangular hyperbolic function along with the two sets of chronaxie values from those two papers (which are in remarkable agreement) to estimate the change in $F_{pulse_{hm}}$ that would be expected from reducing the train duration from 1 s to 0.5 s. The median $F_{pulse_{hm}}$ in the vehicle condition of the current study was 27.1 pulses s$^{-1}$ at a train duration of 1.0 s; the estimated and simulated

value at a train duration of 0.5 s for the vehicle condition is 37.6 pulses s$^{-1}$. We then set $F_{pulse_{hm}}$ for the upper (MFB) limb of the convergence model (Fig 11) to a value typical of eICSS studies (∼77). The graphs in the upper row of Fig 12 were produced by setting the MFB drive to a level equivalent to an optical pulse frequency of 40 pulses s$^{-1}$ (a bit above the estimated $F_{pulse_{hm}}$ value for an oICSS train duration of 0.5 s). This yields the green reward-growth curve shown in the upper-left graph in Fig 12. Asymptotic reward intensity is quite low. One reason for this is that the lower-limb $F_{pulse_{hm}}$ value is well within the roll-off range of the assumed frequency-following function. Another is that frequency-following fidelity in the MFB neurons that generate transsynaptic excitation of the midbrain dopamine neurons is poorer than in neurons that produce the rewarding effect of electrical MFB stimulation [80]. The magenta curve is the sum of the outputs of the upper (cyan curve) and lower (green curve) limbs.

The middle graph in the upper row of Fig 12 shows the simulated effect of dopamine-transporter blockade, with $K_{da}$ set to 1.4125 in the simulation (estimated value in the current study: 1.3951). The simulated effect of the drug shifts the green reward-intensity curve leftwards. Due to the improved frequency-following fidelity in the drug condition, this boosts the upper

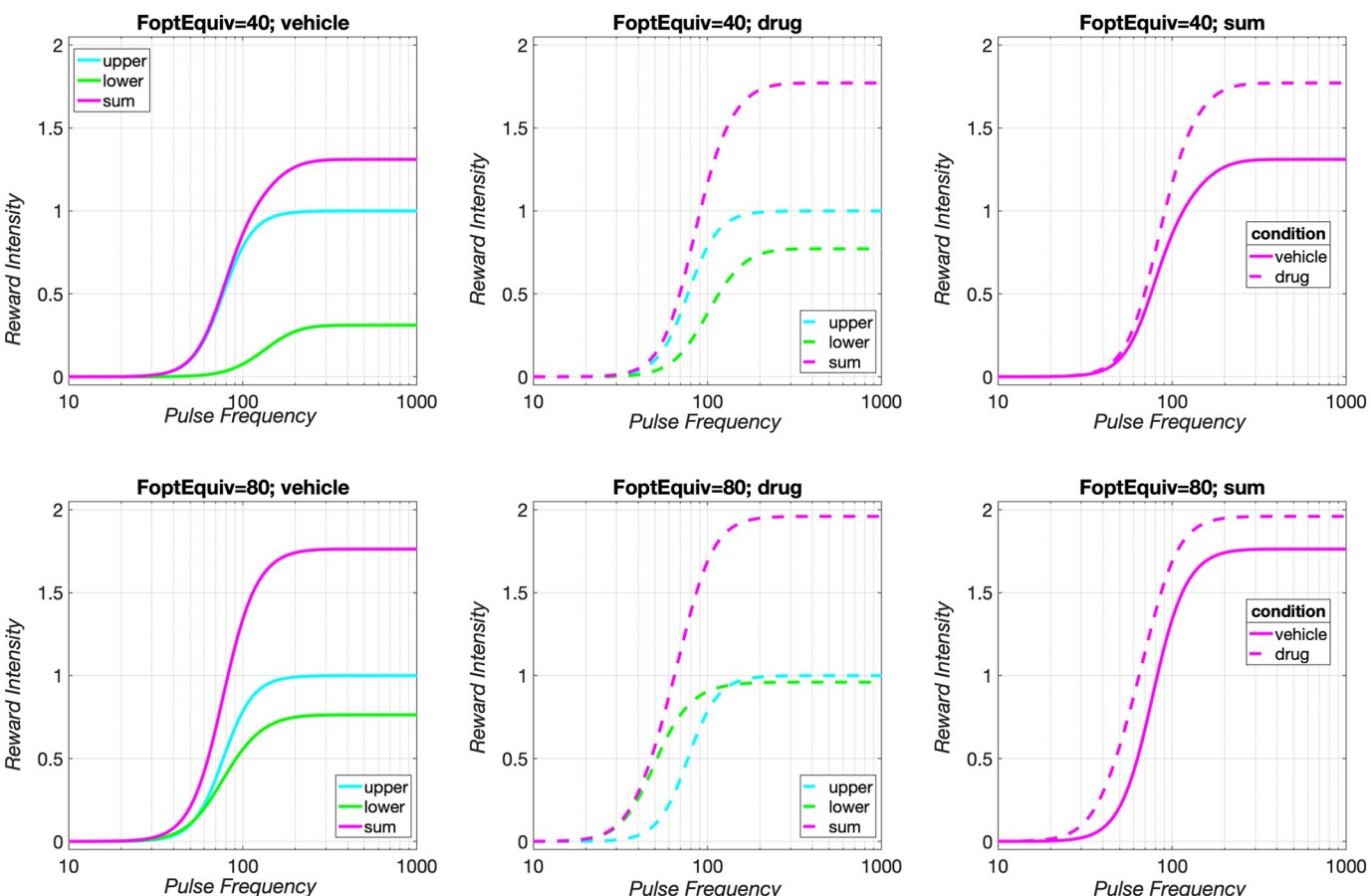

**Fig 12. Reward growth in the convergence model.** The green curves show the growth of reward intensity in the dopaminergic limb of the convergence model (Fig 11, the cyan curves show the growth of reward intensity in the MFB limb, and the magenta curves show the sum of the reward-intensity values in the two limbs. These plots are semi-logarithmic versions of the double-logarithmic plots in Figs 1 and 10. In the upper row, the strength of the MFB drive is equivalent to optical stimulation more than sufficient to generate half-maximal reward intensity, whereas in the lower row, the MFB drive is equivalent to the strongest optical stimulation employed in the present study. The effect of increased tonic-dopamine signaling is presumed to arise from decreases in subjective effort costs or the value of alternate activities but not from upwards rescaling of reward-intensity. Thus the vertical asymptotes of the cyan curves are not altered by the drug.

asymptote of both the dashed green curve for the lower limb and the dashed magenta curve for the summated output of the two limbs. As a result, the reward mountain shifts rightward along the price axis (S35 Fig). (In the simulations, the correction for changes in frequency-following fidelity removes this component of the shift along the price axis, as shown by the dashed cyan line in the bar graph in S35 Fig. The remaining component is tiny and would have been too small to measure given the noise in the behavioral data).

The left shift of the green curve along the pulse-frequency axis in the middle upper panerl of Fig 12 is insufficient to overtake the cyan curve for the upper (MFB) limb of the model. Thus, the summated curve for the drug condition (dashed magenta lines in the upper middle and upper right panels) is not shifted laterally with respect to the summated curve for the vehicle condition (solid magenta lines). As a consequence, the simulated reward mountain does not shift along the pulse-frequency axis (S35 Fig). Thus, the simulated output obtained using moderate MFB drive on the dopamine neurons replicates the eICSS findings.

What would happen if the MFB drive were much stronger? We repeated the simulations setting the MFB drive to the equivalent of an optical pulse frequency of 80 pulses s$^{-1}$, around the highest value tested in most of the subjects in the current study. The results are shown in the lower row of Fig 12. The reward-growth curve for the lower (dopaminergic) limb of the model (green curve) is now shifted leftwards in the vehicle condition (lower-left panel) and rises to a higher asymptote. From its enhanced starting position along the abscissa, the simulated reward-growth curve for the lower limb in the drug condition is now able to overtake the reward-growth curve for the upper limb, shifting the dashed green curve for the lower limb (middle panel, bottom row) to the left of the dashed cyan curve for the upper limb. As a result, the summated curve for the drug condition (dashed magenta lines, bottom row) is displaced somewhat to the left of the summated curve for the vehicle condition (solid magenta lines, bottom row). This shift drags the reward mountain a short distance down the pulse-frequency axis (S36 Fig). Thus, given transsynaptic MFB drive that is sufficiently strong to match extremely intense, direct, optical stimulation, some shift along the pulse-frequency axis is predicted.

This dependence of the output of the convergence model on parameter values is of interest. Although failure to observe shifts along the pulse-frequency axis was the most common result of the experiments in which the eICSS reward-mountain was measured under the influence of drugs that alter dopaminergic neurotransmission (27 of 32 cases reported in [34, 39, 46, 47]), it is not the only result. Reward mountains obtained from three subjects in the cocaine study [39] showed fairly substantial, reliable, shifts along the pulse-frequency axis. (The shift was marginally reliable in a fourth subject when tested initially but disappeared upon re-test.) Although no subject in the GBR-12909 [34] or pimozide [47] studies showed such shifts, one subject in the AM-251 study [46] did. Could variation in the strength of MFB drive on the dopamine neurons explain these apparently discrepant findings?

Taken together, the results of the simulation performed at two levels of MFB drive on the dopamine neurons cast the convergence model in a promising light. With parameter values drawn from the available empirical data (and some as-yet unavoidable generalization from eICSS data to the oICSS case), it can account both for the dominant result of the empirical studies (Fig 12, upper row, S35 Fig) as well as for the few systematic deviations from the general pattern. The emergence of such deviations depends on the strength of the MFB drive and the position of the curve for the upper limb along the abscissa as well as on the frequency-following fidelity of both limbs and the fibers that connect them. Thus, a readily testable prediction of the convergence model is that shifts of the eICSS reward mountain in response to perturbation of dopaminergic neurotransmission should be more common when longer train durations are employed. (Lengthening the train duration allows for better frequency-following

fidelity in the lower limb as well as in the link that relays MFB drive to the dopamine neurons.) Similarly, higher drug doses should increase the prevalence of such shifts.

The convergence model avoids a puzzling feature of the series-circuit model: the apparent imposition of a processing bottleneck in brain-reward circuitry. Large, myelinated, fast-conducting fibers are expensive metabolically, and they take up valuable neural real estate. Why funnel the output of such a costly, high-bandwidth pathway exclusively through lower bandwidth dopamine neurons?

In contrast to the convergence model, the series-circuit model does not account for the combined data from the eICSS and oICSS reward-mountain studies entailing pharmacological alteration of dopaminergic neurotransmission. In the series-circuit model, any manipulation that changes phasic dopamine release, or the post-synaptic consequences of this release, should shift the reward mountain along the pulse-frequency axis. Thirty-three of the 38 subjects of the five eICSS and oICSS studies entailing measurement of the reward mountain under pharmacological challenges argue otherwise.

**A new home for previously orphaned findings?**    The convergence models can account for a heretofore unexplained discrepancy between dopamine release and self-stimulation performance. In rats performing eICSS of the MFB, Cossette and colleagues traded off the stimulation current against the pulse frequency [80]. As expected on the basis of the counter model, increases in pulse frequency required that the current be reduced in order to hold behavioral performance constant. In accord with a more detailed study of the frequency response of the MFB neurons subserving eICSS [8], the required current continued to decline as the pulse frequency was increased beyond 250 pulses s$^{-1}$. In contrast, stimulation-evoked dopamine release in the nucleus accumbens, monitored by means of fast-scan cyclic voltammetry, increased very little, or not at all, as the pulse frequency was increased from 120 to 250 pulses s$^{-1}$ and generally declined when the pulse frequency was increased further to 1000 pulses s$^{-1}$. In the series-circuit model, midbrain dopamine neurons relay signals from the directly activated MFB neurons to the behavioral final common path. Thus, the trade-off between the induced firing frequency and the number of directly activated MFB neurons recruited by the current should be manifested faithfully in the stimulation-induced release of dopamine. It was not. The discrepancy between the trade-off functions for eICSS and dopamine release is not explained by the series-circuit model but is readily accommodated by the convergence model.

Huston and Borbély documented rewarding effects produced by electrical stimulation of the lateral hypothalamic level of the MFB in rats that had undergone near-total ablation of telencephalic structures [84, 85]. Although the major telencephalic terminal fields of the midbrain dopamine neurons had been damaged heavily, the rats learned to perform simple movements to trigger delivery of the electrical stimulation. The series-circuit hypothesis leaves these data unexplained, but the convergence model could accommodate them. For example, hypothalamic or thalamic neurons that survived the ablations might relay reward-related signals to brainstem substrates of the behavioral final common path via MFB fibers, as Huston proposed.

Johnson and Stellar [86] made large, bilateral, excitotoxic lesions in the nucleus-accumbens (NAC) and ventral pallidum (VP) of rats that had been trained to perform eICSS of the lateral hypothalamic level of the MFB. The series-circuit model predicts that such extensive damage to a key dopaminergic terminal fields should have greatly reduced the rewarding effectiveness of the electrical stimulation. It did not. The authors concluded that "while the NAC and VP have been shown to be important for various kinds of reward," . . . they do "not appear to be critical for the expression of ICSS reward. It may be the case that the ICSS reward signal is processed downstream from the NAC and VP and is therefore unaffected by total destruction of

either structure." That conclusion contradicts the series-circuit hypothesis but it is perfectly compatible with, and indeed anticipates, the convergence model presented here.

## The essential role of modeling and simulation

It is common in behavioral neuroscience to make predictions and assess results on the basis of binary, directional classification. The effect of a manipulation either does or does not meet a statistical criterion. If it does, the value of the output variable is said to go up or go down. The numeric values of the input and output variables typically matter little.

The reward-mountain model is more ambitious. It makes predictions about the form of the relationships between the variables. As is so often the case in biology, the relationships are non-linear. There are ranges of an input variable, such as the pulse frequency, over which an output, such as time allocation, changes little and other ranges over which the output changes a lot. Thus, the snide answer to the question of whether an output will change is "it depends," and the serious answer entails specifying this dependency in terms of functional forms and their parameters. The numeric values matter.

Albeit in a modest way, the reward-mountain model confronts the problem of convergent causation, the fact that although behavioral output is highly constrained by the relatively small number of muscles, joints and degrees of freedom in an animal body, a very large number of neural controllers have access to the behavioral output. Thus, different causes can produce the same behavioral output. For example, a given level of reward pursuit may ensue from prior contact with a large, but expensive, reward or a small, inexpensive one. The reward-mountain model distinguishes such cases by tying reward pursuit to both the strength and costs of reward, and it goes beyond binary and directional classification of effects by specifying the functions that map these physical quantities into the subjective values that determine goal selection and behavioral allocation.

The reward-mountain model is quantitatively deterministic: given inputs consisting of reward strengths and costs, it outputs time-allocation values. This output changes in specified, lawful ways when internal parameters, such as the scaling of reward intensities, are perturbed by manipulations, such as drug administration. A virtue of such an approach is that it requires explicit statement of assumptions. A shortcoming, perhaps, is that the specification of the numerous, unavoidable assumptions tends to elicit skepticism that may be eluded by verbal formulations that allow assumptions to remain unstated and implicit. In our view, it is best to make our ignorance manifest so as to incite ourselves and our colleagues to reduce it.

Another sense in which we believe the quantitative specification of the model to be important concerns the often-hidden perils of relying on verbal reasoning alone to predict the behavior of systems embodying multiple, interacting, non-linear components [39, 53]. Verbal reasoning is not up to this task. Instead, it requires careful formal, quantitative specification and demonstration of feasibility via simulation, which can reveal lacunae of which the modeler may have been unaware and generate interesting and unexpected results that inspire new experiments. Perhaps most important is that simulation reveals how the model works in a way that verbal descriptions do not capture, and it provides a strong test of whether, in principle, the model actually does what its designers have intended. In the accompanying Matlab$^{\circledR}$ Live Script, we provide the code for performing simulations of the reward-mountain model. We invite the reader to experiment with the code and thereby critique the model.

The simulations lead us to a view rather different than the one attributed to Otto von Bismark concerning the making of democratic laws and the fabrication of sausages. We believe that when modeling decisions about goal selection and pursuit, very close attention should

be paid to "how the sausage was made." For example, it was only by detailed analysis of the internal workings of the convergence model that we came to appreciate the plausibility of its counter-intuitive prediction: stability of the reward-mountain along the pulse-frequency axis under moderate MFB drive in the face of drug-induced enhancement of dopamine signaling.

Diagrams such as Fig 11 include a large number of components and may invite the viewer to imagine William of Ockham turning in his grave. That said, parsimony entails the jettisoning of *superfluous* entities. We invite the reader to identify which of the components in the models can be tossed overboard without loss of explanatory power. Can the data from previous eICSS experiments and the present oICSS experiment be explained more simply? If we had found an affirmative answer to this question we would have implemented it. Indeed, we see the models discussed here as over-simplified rather than over-complicated. For example, they say nothing about fundamental matters such as the functional specialization of dopamine sub-populations, how signals from the "milieu interne" modulate the decision variables, which operations are performed prior to the recording of key quantities in memory or after their retrieval, or how the subjects learn and update the reward intensities and costs that determine their behavioral allocation. Nonetheless, we argue that the combination of the modeling, simulation and empirical work provides a new perspective on the structure of brain-reward circuitry while challenging a long-established view.

## Finding the MFB substrate

The notion that non-dopaminergic neurons with myelinated axons predominate in the directly stimulated substrate for eICSS of the MFB was proposed in the 1980s on the basis of behaviorally derived estimates of conduction velocity, recovery from refractoriness, and direction of conduction; [10–14, 18]. A 1974 review includes a suggestion that the directly stimulated substrate may be non-dopaminergic [87]. The discrepancy between the behaviorally effective range of pulse frequencies and the frequency-following fidelity of catecholatminergic neurons was also noted during the 1970s [88]. Since that time, little progress has been made toward identifying the directly activated neurons responsible for eICSS of the MFB, although electrophysiological recordings show the rough location of some somata that give rise to fibers with properties compatible with the psychophysically based characterization [89–91]. One likely reason is that the series-circuit model relegates these neurons to a subsidiary role that has thus inspired little empirical investigation: on that view, the MFB fibers merely provide an input, likely one of many [92, 93], to the dopamine neurons ultimately responsible for the rewarding effect.

The convergence model elevates the status of the directly activated neurons subserving the rewarding effect of MFB stimulation. This model asserts that multiple, partially parallel, neural circuits can generate reward and that the dopamine neurons do not constitute an obligatory stage in the final common path for their evaluation and pursuit. From that perspective, it is important to intensify the search for the limb(s) of brain reward circuitry that may parallel the much better characterized dopaminergic pathways. Application of modern tracing methods that integrate approaches from neuroanatomy, physiology, optics, cell biology and molecular biology (e.g., [94]) may well achieve what application of the cruder, older tools failed to accomplish. The detailed psychophysical characterization of the quarry that has already been achieved, particularly the evidence for myelination and axonal trajectory [10–14, 89], can guide the application of such methods.

## Potential implications of parallel channels in brain-reward circuitry

Remarkable success has been achieved in developing tools for specific excitation or silencing of dopaminergic neurons, measuring the activity of these neurons, mapping the circuitry in

which they are embedded, and categorizing different functional dopaminergic subpopulations. The neurobiological study of dopamine neurons has been coupled to learning theory, neural computation, and psychiatry, with longstanding application in the study of addictive disorders [6, 16, 95, 96] and emerging linkage to the analysis of depression [97, 98]. Perhaps the brilliance of these successes has obscured the possible roles played by other neurons and circuits in the functions in which dopaminergic neurons have been implicated. We propose that the network in which dopamine neurons are embedded is not the sole source of input to the behavioral final common path for the evaluation and pursuit of rewards. What roles might the alternative sources play in behavioral pathologies and behaviors essential to well being? Addressing that question requires that the existence of such networks be appreciated and addressed, their constituents identified, and their function understood.

## Conclusion

The reward-mountain model was developed to account for data from eICSS experiments in which rats worked for rewarding electrical stimulation of the MFB. At the core of the model is a logistic reward-growth function that translates the aggregate impulse flow induced by the electrode into a neural signal representing the intensity of the reward. On the basis of the direction in which a drug shifts the reward mountain within a space defined by the strength and opportunity cost of reward, the model distinguishes drug actions at, or prior to the input to the reward-growth function from actions at, or beyond, the output. Bidirectional perturbations of dopaminergic neurotransmission have acted selectively in the latter manner, either by rescaling the output of the reward-growth function or by altering other valuation variables, such as subjective reward costs or the attraction of alternate activities: Dopamine-transporter blockers [34, 39] and a dopamine-receptor antagonist [47] shifted the mountain along the axis representing opportunity cost and not along the axis representing pulse-frequency.

The present study demonstrates that the reward-mountain model also provides a good description of how the strength and opportunity cost combine to determine the allocation of behavior to pursuit of direct, optical activation of midbrain dopamine neurons. The results argue that as in the case of eICSS, a logistic-like function with independent input-scaling and output-scaling parameters translates the neural excitation induced by the stimulation into the intensity of the rewarding effect. Unlike the case of eICSS, augmentation of dopaminergic neurotransmission by dopamine-transporter blockade acts as if to rescale the input to the reward-growth function: the reward mountain was shifted along the pulse-frequency axis. In one sense, this is not surprising: by boosting the peak amplitude of stimulation-induced, dopamine-concentration transients in terminal fields, the drug should be expected to reduce the pulse frequency required to generate a reward of a given intensity. However, this result challenges a longstanding model of brain-reward circuitry.

According to the series-circuit model, the rewarding effect of electrical MFB stimulation arises from the activation of highly excitable, non-dopaminergic axons that provide direct or indirect synaptic input to midbrain dopamine neurons; the rewarding effect arises from the transsynaptic activation of these dopamine neurons. The current results show that when these dopamine neurons are activated directly, dopamine-transporter blockade shifts the reward mountain along the pulse-frequency axis. We argue that dopamine-transporter blockade should produce a similar shift in the position of the reward mountain when the dopamine neurons are activated transsynaptically. However, specific perturbation of dopaminergic neurotransmission has almost always failed to shift the reward mountain along the pulse-frequency axis when electrical stimulation of the MFB is substituted for optical stimulation of the

midbrain dopamine neurons. Thus, the series-circuit model cannot readily accommodate the results of both the eICSS and oICSS experiments.

We propose that alternatives to the series-circuit model be explored. In the one sketched here, the reward signal carried by the MFB axons runs parallel to the reward signal carried by the midbrain dopamine neurons prior to the ultimate convergence of these two limbs of brain-reward circuitry onto the final common path for the evaluation and pursuit of rewards. This proposal can accommodate findings unexplained by the series-circuit model and suggests a research program that could complement the work that has so powerfully and convincingly implicated dopaminergic neurons in reward.

## Materials and methods

### Subjects

Seven TH::Cre, male, Long-Evans rats weighing $\sim$350 g at the time of surgery, served as subjects. Animals were obtained from a TH::Cre rat colony established from sires generously donated by Drs. Ilana Witten and Karl Deisseroth. Upon reaching sexual maturity, animals were housed in pairs in a 12 h -12 h reverse light-cycle room (lights off at 8:00 AM). The rats were housed singly following surgery.

### Ethics statement

All experimental procedures were approved by the Concordia University Animal Research Ethics Committee (Protocol #: 30000302) and conform to the requirements of the Canadian Council on Animal Care.

### Surgery

Midbrain dopamine neurons were transfected with the light-sensitive cation channel, *channelrhodopsin 2* (ChR2), fused to the reporter protein, *enhanced yellow fluorescent protein* (eYFP). The construct was delivered by means of a Cre-dependent Adeno-Associated Viral vector (AAV5-DIO-ChR2-EYFP, University of North Carolina Viral Vector Core, Chapel Hill, NC). The virus was injected bilaterally ($\pm$0.7 mm ML), at a volume of 0.5 $\mu$l, at three different DV coordinates (-8.2, -7.7 and -7.2 mm) and two different AP coordinates (-5.4 and -6.2 mm), to yield a total volume of 3.0 $\mu$l per hemisphere. Optical-fiber implants, with a core diameter of 300 $\mu$m, were aimed bilaterally at the VTA at a 10$^\circ$ angle (AP: -5.8, DV: 8.02 or 8.12, ML: $\pm$ 0.7 mm). Anesthesia was induced by an i.p. injection of a Ketamine-xylaxine mixture (87 mg/kg, 13 mg/kg, Bionicle, Bellville, Ontario and Bayer Inc., Toronto, Ontario, respectably). Atropine sulfate (0.02-0.05 mg/kg, 1 mL/kg, Sandoz Canada Inc., Quebec) was injected s.c. to reduce bronchial secretions, and a 0.3 mL dose of penicillin procaine G (300 000 IU/ml, Bimeda-MTC Animal Health Inc., Cambridge, Ontario) was administered SC, as a preventive antibiotic. "Tear gel" (1% w/v, 'HypoTears' Novartis) was applied to the eyes to prevent damage from dryness of the cornea. Anesthesia was maintained throughout surgery by means of isoflurane ($1 - 2.5\% + O_2$). The head of the rat was fixed to the stereotaxic frame (David Kopf instruments, Tujunga, CA) by means of ear bars inserted into the auditory canal and by hooking the incisors over the tooth bar. Bregma and Lambda were exposed by an incision of the scalp. Three blur holes were drilled in the skull over each hemisphere (AP: -5.4, -5.8, and -6.2 mm; ML:$\pm$ 0.7,$\pm$ 2.08,$\pm$ 0.7, respectively). A 28 gauge injector was loaded with the viral vector. Six 0.5 $\mu$L boli of the virus-containing suspension were infused into each brain hemisphere at the following coordinates: AP: -5.4 and -6.2 mm; ML: $\pm$ 0.7 mm; DV: -8.2, -7.7 and -7.2 mm. Infusions were performed at a rate of 0.1 $\mu$L per minute using a precision pump

(Harvard Instruments) and a 10 $\mu$L Hamilton syringe (Hamilton Labaoratory prodcuts, Reno, NV). To allow for diffusion, the injector was left in place for ten minutes following each infusion. Optical-fiber implants with a 300 $\mu$m, 0.37 numerical-aperture core were constructed following the methods described by Sparta et al. [99]. Optical fibers were aimed bilaterally at the VTA at a 10˚ angle. The implants were placed at two different DV coordinates to increase the chances of placing the tip of at least one of the optical fibers directly over the neurons that support optical self-stimulation (AP −5.8 mm; ML ± 0.7 mm; DV −8.02 and -8.12 mm). The optical implants were anchored to the skull by means of stainless steel screws and dental acrylic. Gelfoam™ (Upjohn Company of Canada, Don Mills, Ontario) was used to fill the holes in the skull and promote healing. Buprenorphine (0.05ʻmg/kg SC, 1 mL/kg, RB Pharmaceuticals Ltd., Berkshire, UK) was used as a post-surgery analgesic. The rats were housed singly in the animal care facility for over five weeks to allow for surgical recovery and to achieve appropriate expression and distribution of the protein product of the ChR2-EYFP construct.

## Apparatus

The operant chambers (30 × 21 × 51 cm) had a mesh floor and a clear Plexiglas front equipped with a flashing light located 10 cm above the floor mesh, and a retractable lever (ENV–112B, MED Associates) mounted on a side wall. A 1 cm light was located 2 cm above the lever and was activated when the rat depressed the lever. A blue DPSS laser (473 nm, Shanghai Lasers and Optics Century Co. or Laserglow Technologies, Toronto, ON) was mounted on the roof of each box. The laser was connected to a 1 × 1 FC/M3 optical rotary joint (Doric lenses, Quebec, Canada) by means of a laser coupler (Oz Optics Limited, Ottawa, ON, or Thorlabs, Inc., Newton, New Jersey, USA) and fiber-optic patch cords. Robust, custom-built, optical-fiber patch cords designed for rats [100] were used to attach the implants in the animal's head to the 1 × 1 FC/M3 optical swivel so as to allow the rat to move without tangling the cable. Experimental control and data acquisition were handled by a personal computer running a custom-written program ("PREF") developed by Steve Cabilio (Concordia University, Montreal, QC, Canada). The temporal parameters of the electrical stimulation were set by a computer-controlled, digital pulse generator. Stimulation consisted of 1 s trains optical pulses, 5 ms in duration.

## Drug

GBR-12909 was dissolved in 0.9% saline at a volume of 10 mg/ml. The pH of the solution was adjusted to 5± 0.1 with 0.1M NaOH.

## Self-stimulation screening and initial training

Each animal underwent two to three screening sessions in which only one of the optical implants was attached to the laser. Optical power was measured by means of an optical power meter (PM100D, Thorlabs Inc., Newton, New Jersey, USA, modified to reduce spurious output noise) and adjusted through trial and error for each rat to elicit robust oICSS behavior (30-60 mW, measured at the tip of the patch-cord with the laser operating in continuous-wave mode). Animals were trained by means of the successive approximation procedure to depress the lever to receive optical stimulation (a 1 sec train of 5 ms optical pulses at 80 pulses per second (pps). After the rats had learned to lever press, they were allowed to work for the optical reward, on a continuous-reinforcement schedule, during two 15 min trials. The total number of presses was recorded. Both of the implants were tested under these conditions; the implant that yield the larger number of presses was used for the rest of the experiment.

## Training in preparation for measurement of the reward mountain

The rats received further training to prepare them for measurement of the reward mountain (S37 Fig). First, the rats were trained to perform a new reward-procuring response, holding down the lever rather than simply pressing it briefly. This new response was rewarded according to a cumulative-handling-time schedule of reinforcement [101], which delivers a reward when the cumulative time the lever has been depressed reaches an experimenter-defined criterion called the "price" of the reward. In this sense, the price corresponds to what economists call an opportunity cost. To earn a reward, the rats did not have to hold down the lever continuously until the price criterion was met; they could meet the criterion by means several bouts of lever holding separated by pauses. The onset and offset times of each bout of lever depression were recorded.

Next, the values of one or both independent variable (pulse frequency, price) were varied sequentially from trial to trial, thus traversing the independent-variable space along a linear trajectory. Such a traversal is called a "sweep." Three types of sweeps were carried out. Frequency sweeps were carried out at a fixed price (initially 1-2 s). These sweeps consisted of 10 to 12 trials during which the rat had the opportunity to harvest as many as 60 rewards (except for rat BeChr19, who was allowed to harvest a maximum of 30 rewards per trial due to the unusual effectiveness of optical stimulation in this subject). Each reward was followed by a 2 s *Black Out Delay* (BOD) during which the lever was disarmed and retracted, and timing of the trial-duration was paused. The pulse frequency during the first two trials was set to yield maximal reward-seeking behavior. The first trial was considered a warm-up trial and was excluded from analysis. From the second trial onwards, the rewarding stimulation was decreased systematically from trial to trial by decreasing the pulse frequency in equal proportional steps. The range of tested pulse frequencies was selected as to drive reward-seeking behavior from its maximal to its minimal value in a sigmoidal fashion (S37B). Every trial was preceded by a 10 s *Inter-Trial Interval* (ITI) signaled by a flashing light. During the last 2 s of this period rats received priming stimulation consisting of a non-contingent, 1 s stimulation train, delivered at the maximally rewarding pulse frequency. Rats performed one frequency sweep per session, consisting of ten trials (except for subject BeChr19 who was tested on 12 trials per frequency sweep).

When the rats showed consistent performance across trials and sessions in the frequency-sweep condition, we introduced price sweeps into the training sessions. In the price-sweep condition, the pulse frequency was kept constant at the maximal value for each rat, but the cumulative work time required to harvest the reward (i.e. the price of the reward) was increased systematically across trials. The price was the same on the first two trials of each sweep. As in the frequency-sweep condition, the first trial was considered a warm-up trial and was excluded from analysis. Starting at the second trial of the sweep, prices were increased by equal proportional steps across trials. The prices were set by trial and error so as to yield a sigmoidal transition between maximal and minimal reward-seeking behavior as a function of price (S37B Fig). Trial duration was set so as to allow the rats to harvest a maximum of 60 rewards per trial (30 in the case of BeChr19). The BOD, ITI, and priming were the same as in the frequency sweep. During price-sweep training sessions, the rats also performed a frequency sweep. The order of the sweeps was randomized across sessions.

Radial sweeps were incorporated when performance on the price sweeps appeared stable. Along radial sweeps, the pulse-frequency was decreased and the price was increased concurrently across trials. Thus, frequency sweeps run parallel to the pulse-frequency axis in the independent-variable space, price sweeps run parallel to the price axis, and radial sweeps run diagonally. The radial sweeps were composed of ten trials; the first trial served as a warm-up

and was identical to the second trial. From the second trial onwards, both pulse frequency and price were varied in equal proportional steps so as to yield a sigmoidal decrease in reward-seeking behavior over the course of the sweep (S37B Fig). The trajectory of the vector defined by the tested pulse frequencies and prices was aimed to pass as close as possible to the point defined by the estimated values of the $F_{pulse_{hm}}$ and $P_{obj_e}$ location parameters (see model fitting section). This required fitting the mountain model to preliminary data from each rat and adjusting the pulse frequencies and prices tested along the radial sweep accordingly for the following session. Pulse-frequency and price sweeps were also performed in each session during this phase of testing. The order of presentation of the three different sweeps was random across sessions, and the BOD, ITI, and priming parameters were the same on all trials. Rats were considered ready for drug-test sessions when reward-seeking behavior declined sigmoidally and consistently along all three sweeps and the trajectory of the radial sweep in the independent-variable space passed close to the point defined by the estimated values of $F_{pulse_{hm}}$ and $P_{obj_e}$.

## Effects of GBR-12909 on the reward mountain

Rats received i.p. injections 90 min prior to behavioral testing. Vehicle (2.0 ml/kg) was administered on Mondays and Thursdays and GBR-12909 (20 mg/kg) on Tuesdays and Fridays. In each session, rats performed a frequency, a price, and a radial sweep in random order. Each sweep consisted of ten trials each (except for the frequency sweep for rat BeChr19, which consisted of 12 trials). The duration of each trial was set so as to allow rats to harvest a maximum of 60 rewards per trial (except for rat BeChr19, who was allowed to harvest a maximum of 30 rewards per trial due to his unusual proclivity to work for very high opportunity costs). Wednesdays and weekends were used as drug elimination days: no testing was conducted on these days, and the rats remained in the animal care facility. Ten vehicle and ten drug sessions were conducted with each rat.

## Calculation of time allocation

The raw data were the durations of "holds" (intervals during which the lever was depressed by the rat) and "release times" (intervals during which the lever was extended but not depressed by the rat). Total work time included 1) the cumulative duration of hold times during a trial, and 2) release times less than 1 s. The latter correction was used because during very brief release intervals, the rat typically stands with its paw over or resting on the lever [101]. Therefore, we treat these brief pauses as work and subtract them from the total release time. Corrected work time for a given trial was defined as the sum of the corrected hold times, and leisure time was defined as the sum of the corrected release times. The dependent measure was time allocation (TA), the ratio of the corrected work and leisure times.

## Model fitting and comparisons

A separate TA calculation was performed for each *reward encounter*: the time between extension of the lever and completion of the response requirement (when cumulative work time equals the set price). The primary datasets thus consisted of the TA values for the reward encounters that occurred during all trials and sessions run following administration of the drug or vehicle. These primary datasets were then resampled 250 times with replacement [39, 102].

**Fixed parameters.** The mountain model and the fitting approach have been described in detail elsewhere [38, 39]. Two versions of the extended reward-mountain model [38] were fit to the present data. Both models include four fixed parameters, two describing the subjective-

price function [40] and two describing the frequency-following function [8]. The subjective-price function maps the objective price of the reward into its subjective equivalent. Here, we used the form and parameters of obtained for this function in a study of eICSS of the MFB [40]. The frequency-following function maps the optical pulse frequency into the induced frequency of following in the midbrain dopamine neurons. The function we used here was of the same form as the one described in a study of eICSS of the MFB [8], but different parameter values were required to accommodate the differences between the frequency responses of optically stimulated ChR2-expressing midbrain dopamine neurons and electrically stimulated MFB neurons subserving eICSS. For details, please see the section entitled "Parameters of the frequency-following function for oICSS" in the S1 File.

**Fitted parameters.** The first ("standard") model includes six fitted parameters. The location parameters $\{F_{pulse_{hm}}, P_{obj_e}\}$ position the mountain along pulse-frequency and price axis, respectively (Fig 1B), the $a$ and $g$ parameters determine the slope of the mountain surface, and the $T_{min}$ and $T_{max}$ parameters determine its minimum and maximum altitudes, respectively. The second ("CR") model includes an additional parameter, $CR$, that estimates the contribution of conditioned reward [34, 39]. This parameter selectively increases time allocation to pursuit of weak, inexpensive rewards, thus providing a more accurate fit when the lever and/or the act of depressing it become potent secondary reinforcers.

**Common versus treatment-dependent parameters.** Models containing many parameters can prove excessively flexible, and fits employing them may fail to converge. We restricted the flexibility of the models by fitting common values of $T_{min}$ and $T_{max}$ to the data from the two treatment conditions {vehicle, drug}. The rationale is that the factors causing $T_{min}$ to deviate from zero and $T_{max}$ to deviate from one tend to be be common across vehicle- and drug-treatment conditions. The main experimental question posed concerns the effect of the drug on the location parameters. Thus, these two parameters $\{F_{pulse_{hm}}, P_{obj_e}\}$ were always free to vary across treatment conditions. Variants of both the standard and CR models were produced in which all, some, or none of the $a$, $CR$, and $g$ parameters were common across the two treatment conditions. Thus, twelve models were fit: four variants of the standard model

1. *a* free, *g* common

2. *g* free, *a* common

3. both *a* and *g* free

4. neither *a* nor *g* free

and eight variants of the CR model (the same combinations as for the standard model, but with *CR* either free or common).

All 12 models were fit to each of the 250 resampled datasets using a procedure developed by Kent Conover, based on the nonlinear least-squares routine in the MATLAB optimization toolbox (the MathWorks, Natick, MA). Mean values for each parameter were obtained by averaging the 250 estimates. Confidence intervals (95%) were estimated by excluding the lowest and the highest 12 values of the 250 estimates. This yields unbiased estimates of the fitted parameters and their dispersions for each subject. The Akaike information criterion [54] was used to select the model that offers the best balance between achieving a good fit and minimizing the number of parameters required to do so. Drug-induced shifts in the location of the 3D structure were considered significant when the 95% confidence interval around the difference between the 250 resampled estimates of the location parameters across drug and vehicle conditions excluded zero (i.e. no difference between conditions).

## Power-frequency trade-off

Following the pharmacological experiment, we explored the effects of systematic changes in optical power and pulse frequency on the number of rewards obtained by each rat. The same subjects were trained to press the lever on an FR-1 schedule to trigger optical stimulation of midbrain dopamine neurons through the same optical implant used in the reward-mountain experiment. We quantified the number of responses emitted in each of a series of nine or ten trials lasting two minutes each. The pulse frequency was decreased systematically across trials: the highest pulse frequency was in effect during the first trial of the series (the "warm-up") and also on the second trial, and data from the first trial was excluded from analysis. Subsequently, the pulse-frequency decreased in equal logarithmic steps from trial to trial. A single pulse-frequency sweep was run in each of five to six sessions per day. In each session, the optical power (measured at the tip of the patch-cord with the laser operating in continuous-wave mode) was set to one of five values (1.87, 3.75, 7.5, 15 or 30 mW for rat Bechr14; 3.75, 7.5, 15, 30, or 60 mW for rats Bechr21, Bechr28 and Bechr29; and 2.5, 5, 10, 20, and 40 mW for rats Bechr19, Bechr26, and Bechr27). The rats were tested under these conditions for five days. The order of presentation of optical powers was determined pseudorandomly for each rat. Each power was presented in a different sequential order across each testing day (i.e. each power was used 1st, 2nd, 3rd, 4th, or 5th at least once across days, but the preceding and/or subsequent tested powers may have been different across test days). To control for carry-over effects across test sessions, rats were given 30-min breaks between each session: following completion of each of the five to six daily sessions, they were taken out of the operant chambers, brought back to their home cage in the animal-care facility, and had free access to food and water for 30 min before resuming with the following test session. During this 30-min break, the lasers were set to continuous operation mode to minimize fluctuation in optical power due to the cooling of the laser.

## Histology

Rats were sacrificed by means of a lethal injection of pentobarbital i.p. After deep anesthesia had been induced, the rats underwent intracardiac perfusion with phosphate-buffered saline and 4% paraformaldehyde chilled to 4˚C. Upon extraction, the brains were cryoprotected in a solution of 4% paraformaldehyde and 30% sucrose for 48 h at 4˚C and transferred to a −20˚C freezer thereafter. The brains were sliced coronally in 40 $\mu m$ sections by means of a cryostat, and mounted in electrostatically adhesive slides (Fisherbrand™ Superfrost™ Plus slides, Fisher Scientific, Pittsburgh, PA). Sections were washed in 0.3% triton in Phosphate-Buffered Saline (PBS) for two minutes, then immersed in 10% donkey serum in PBS for 30 min for blocking. Anti-Tyrosine hydroxylase antibody (MilliporeSigma, AB152) was diluted to 1:500 and incubated overnight at room temperature. The sections were then washed three times for five min in PBS and incubated with Alexa fluor 594 secondary antibody (Jackson Immuno research laboratories A-11012) for two hours at room temperature. The slides were washed three times for five minutes in PBS and coverslipped using Vectashileld with DAPI. Expression of the construct and restriction to TH-positive sites was confirmed using epifluorescence and confocal microscopy.

## Modeling

A Matlab (version 2019b, The Mathworks, Natick, MA) Live Script was used to simulate the output of the reward-mountain model. Models of the neural circuitry underlying eICSS and oICSS were explored by means of the simulations. The Live Script develops the mountain model from first principles, both for eICSS and oICSS. Key experiments assessing the validity of the model are reviewed, and simulated results are compared to empirical ones. The predictions

of the series-circuit model of brain reward circuitry are tested and found to deviate from the empirical results of eICSS studies. An alternate model is proposed and used to simulate the results of eICSS experiments that are not readily explained by the series-circuit model. The text of the Live Script is provided in the Supporting Information along with instructions for downloading and installing the executable code.

## Supporting information

**S1 File.**
(PDF)

## Acknowledgments

Steve Cabilio developed and maintained the experimental-control and data acquisition software used in this study. The experimental-control and data acquisition hardware was designed, built, and maintained by David Munro. Karl Deisseroth and Ilana B. Witten kindly provided TH::Cre$^{+\backslash\text{-}}$ rats to establish our breeding colony. Marie-Pierre Cossette assisted Ivan Trujillo-Pisanty in genotyping and breeding-colony maintenance. The NIMH Chemical Synthesis and Drug Supply Program generously donated the GBR-12909 used in this study. We received valuable advice on microscopy from Chloë Van Oostende (Concordia Centre for Microscopy and Cellular Imaging). Elizabeth E. Steinberg provided helpful guidance concerning oICSS. Andreas Arvanitogiannis, Peter Dayan and Giovanni Hernandez offered much appreciated comments on the manuscript.

## Author Contributions

**Conceptualization:** Ivan Trujillo-Pisanty, Peter Shizgal.

**Data curation:** Peter Shizgal.

**Formal analysis:** Kent Conover, Peter Shizgal.

**Funding acquisition:** Peter Shizgal.

**Investigation:** Ivan Trujillo-Pisanty, Pavel Solis, Daniel Palacios.

**Methodology:** Ivan Trujillo-Pisanty, Peter Shizgal.

**Project administration:** Peter Shizgal.

**Resources:** Peter Shizgal.

**Software:** Kent Conover, Peter Shizgal.

**Supervision:** Peter Shizgal.

**Validation:** Peter Shizgal.

**Visualization:** Kent Conover, Peter Shizgal.

**Writing – original draft:** Ivan Trujillo-Pisanty, Peter Shizgal.

**Writing – review & editing:** Peter Shizgal.

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
