## [Decision Letter · Decision Letter 0]

13 Feb 2020

PONE-D-19-33493

The effect of dopamine transporter blockade on optical self-stimulation: behavioral and computational evidence for parallel processing in brain reward circuitry

PLOS ONE

Dear Dr. Shizgal,

Thank you for submitting your manuscript to PLOS ONE. After careful consideration, we feel that it has merit but does not fully meet PLOS ONE’s publication criteria as it currently stands. Therefore, we invite you to submit a revised version of the manuscript that addresses the points raised during the review process.

While reviewers were impressed by your submission, both provided helpful suggestions and presented questions that should be easily addressable. It is my opinion that addressing these questions and suggestions will significantly improve this submission.

We would appreciate receiving your revised manuscript by Mar 29 2020 11:59PM. To enhance the reproducibility of your results, we recommend that if applicable you deposit your laboratory protocols in protocols.io, where a protocol can be assigned its own identifier (DOI) such that it can be cited independently in the future. For instructions see: http://journals.plos.org/plosone/s/submission-guidelines#loc-laboratory-protocols

We look forward to receiving your revised manuscript.

Kind regards,

Juan M Dominguez, PhD

Academic Editor

PLOS ONE

Journal Requirements:

3. Thank you for stating the following in the Competing Interests/Financial Disclosure* (delete as necessary) section:

PS; Natural Sciences and Engineering Research Council of Canada grants #s RGPIN

308-11,RGPIN-2016-06703; https://www.nserc-crsng.gc.ca

We note that one or more of the authors are employed by a commercial company: OMMax Digital Strategy.

4. Please ensure that you refer to Figure 12 in your text as, if accepted, production will need this reference to link the reader to the figure.

Reviewers' comments:

Reviewer's Responses to Questions

**Comments to the Author**

1. Is the manuscript technically sound, and do the data support the conclusions?

Reviewer #1: Yes

Reviewer #2: Yes

2. Has the statistical analysis been performed appropriately and rigorously? 

Reviewer #1: Yes

Reviewer #2: Yes

3. Have the authors made all data underlying the findings in their manuscript fully available?

Reviewer #1: Yes

Reviewer #2: Yes

4. Is the manuscript presented in an intelligible fashion and written in standard English?

Reviewer #1: Yes

Reviewer #2: Yes

5. Review Comments to the Author

Reviewer #1: Review of PONE-D-19-33493

Summary

To appreciate the significance of this paper one needs to know the relevant background, which I now summarize.

Optogenetic techniques have given new impetus to the study of the neural substrates for positive reinforcement, that is, for the neural signals conveying a message, the memory for which induces goal-oriented behavior. The phenomenon of electrical self-stimulation of the brain induces truly spectacular goal oriented behavior: Rats will work around the clock to obtain direct electrical stimulation of pathways within the medial forebrain bundle, neglecting the pursuit of food, water and sleep in order to obtain to continually obtain this reinforcement.

The medial forebrain bundle contains many projections, running either caudo-rostrally or rostro-caudally, thereby linking loci in the tegmentum and in the forebrain. A question of obvious and fundamental importance has always been, Which of these many projections carry the reinforcing signals produced by the reinforcing stimulation of the reinforcement-relevant axons?

It was possible by psychophysical means to measure mportant electrophysiological and circuit-level properties of these projections. (The psychophysical method obtains trade-off functions that describe combinations of stimulation parameters that produce a constant behavior effect.) The refractory periods, strength-duration characteristics, conduction velocities and orthodromic direction of conduction for the relevant directly stimulated axons were learned by these methods as was the surprising fact that they are distributed fairly homogenously in an appreciable radius around the tip of the stimulating electrode. It was also established that the relevant directly stimulated axons projected rostro-caudally. It was also learned that the magnitude of the remembered reward depends on the total number of action potentials produced in the relevant axons within a 1-second signal-integration period; it does not matter whether that total is produced by exciting relatively few axons many times or many axons a few times.

Electrophysiological studies, mostly conducted by the Shizgal lab, showed that there were directly stimulated axons with the properties required by the psychophysical results. They also showed that these were NOT properties of the axons in dopaminergic projections. In other words, the reinforcement-relevant axons directly excited by electrical stimulation of the MFB are not dopaminergic. However, it was never possible to identify by these means—nor, puzzlingly, by lesions and tract tracing—the neuroanatomical and neurochemical identity of the relevant non-dopaminergic projections.

Another big puzzle was the connection between what was revealed by psychophysical and electrophysiological means and what was revealed by pharmacological means. It was established long ago that the blockade of dopaminergic receptors makes rats unwilling to respond for it. The effect was somewhat similar to reducing reinforcement intensity by reducing either pulse frequency or current; blocking drugs sort of moved the psychometric function (response rate vs stimulation parameter) rightward toward higher values of the trade-off parameter (current or pulse frequency). However, the shift was never large; and as the dose of the blocker increased, the asymptote of the psychometric function decreased very substantially. The logic here is identical to the interpretation of dose curves in pharmacology: a competitive blocker does not put receptors out of play, it merely requires an increased agonist intensity (more agonist) to achieve a given effect. By contrast, a non-competitive blocker that binds covalently, puts receptors out of play (unavailable to the agonist), thereby reducing the achievable performance. A competitive blocker shifts the dose-response curve to the right and does not lower its asymptote, whereas a non-competitive blocker lowers the asymptote and does not shift the curve to the right. The effects of dopamine blockers on the psychometric function look rather more like the effect of a non-competitive variable such as the cost of the reinforcement rather than an effect on the intensity of reinforcement. As the cost of a cookie goes up, our willingness to work for it goes down even though it tastes as yummy as ever, whereas blockade of sugar receptors makes it taste less yummy. No matter how legendary the taste of a find wine, if it costs a billion dollars, we will go for it, even though we may sing its praises to our grandchildren. In Berridge and Robinson terminology, the liking for a preposterously expensive wine once tasted never goes away, but as it grows more costly the willingness to buy it, that is, work for it, goes down. The memory of the wine's intensely pleasing taste—its reinforcing intensity— still makes us want it, but its cost makes us unwilling to work/pay for it. By contrast, if the wine maderizes, so we no longer like it, we will in this case, too, not go for it, but nor will we sing its praises to our grandchildren. After, Shizgal's lab developed and validated the reward-mountain method, they were able to show that the effect of dopaminergic blockade was indeed on a performance/cost variable not on reinforcement intensity. In short, we now know that blockading dopaminergic transmission does not diminish the reinforcing effect of brain stimulation reinforcement; it increases some cost variable to the point where the rats will no longer work for that wonderful reinforcement, just as this reviewer will no longer buy a bottle of Chateau d'Yquem, though s/he remembers with extreme fondness the days when its price was within reason.

The discovery that dopamine blockade increased the cost of brain stimulation reinforcement without altering its intensity explained a profoundly puzzling result from the Gallistel lab: dopaminergic blockade that eliminates responding for brain stimulation reward does not at all block the priming effect the transient effect that determines how much the animal wants the reinforcement. Priming the rat makes it forage intently for remembered brain stimulation reward, but the effect fades rapidly with time (over a minute or so). When a pimozide-treated is primed, it vigorously seeks more of that reinforcement at locations where it was previously obtained or by pressing a lever that previously produced it. But this happens only the first few times the pimozide-treated rat is primed because when it does forage for that well-remembered reinforcement, it rapidly discovers that its cost is too great. Again, the following will, I hope, make this intuitive. Suppose me to be oblivious to the absurd increase in the cost of Chateau d'Yquem since my youth. Suppose further that it has been years since I tasted that ambrosia. Another taste reawakens my wanting (in Berridge and Robinson terminology), but when I run to the store to buy some (forage for it), I am shocked to discover its outrageous cost, and I retreat disappointed. After a few such disappointments, renewed tasting of it, reawaken fond memories, but they no longer causes me to run to the store, because I now have learned its outrageous cost. Thus, tasting (priming) prompts wanting somethingone likes, but not—after a few disappointing discoveries about cost—acting.

Bottom line so far: Electrical self-stimulation of the brain excites rostro-caudally conducting axons that carry the signal that results in a remembered reinforcing effect (after signal integration across time, i.e., train duration, and space, i.e. number of relevant axons excited by each pulse). Given the old (pre-reward mountain) results on dopamineric blockade, it was reasonable to assume that the descending signal in these axons excited an ascending dopaminergic projection that carried the signal to the point of signal-integration and memory formation. However, the reward-mountain results showed that this is not the case. Dopaminergic blockade acts after signal integration and, in all probability, after memory formation. Memory formation is not shown in Shizgal's diagram (fig 2) but without it, of course, the rats could not show behavior directed toward obtaining the remembered effect. The primed rats have not forgotten the reinforcing intensity and where and how it was produced, because when primed again they will seek it there and by those means no matter how much time has elapsed since last they sought it. Thus, the pathways for brain-stimulation reinforcement are nerve-memory pathways, which further broadens their neurobiological interest.

The development of optogenetic methods for directly exciting neuropharmacologically and neuroanatomically specified projections within the MFB was by far the most important development in the last decade or so in this area. We now know that directly stimulating a dopaminergic projection in the MFB can produce a reinforcing effect. Does this imply that the earlier conclusion that the reinforcing signal from electrical stimulation of the projections in the MFB is not transmitted to the site of signal integration by a dopaminergic projection? This is the question posed by the current paper. Its importance can only be understood by those familiar with the background I have just sketched. That background is not widely known because the study of electrical self-stimulation almost died out prior to the development of optogenetic methods, which development has reawakened interest in the neural substrate for the powerful and effects of traditional reinforcers (food, water and sex) on goal-directed behavior.

Importance of the results

While I have not followed this literature in many years, this paper, to my knowledge, is the first to bring state-of-the-art psychophysical methods to bear on the behavior produced by rewarding optical stimulation of dopaminergic projections. And, it does so to very telling and, most unexpected effect. Because, as I mentioned at the outset, reinforcing electrical stimulation produces such spectacularly vigorous, dedicated and prolonged behavior directed to obtaining ever more of it, I was stunned to learn that optical stimulation can produce a much more intense reinforcing effect.

Most readers will be even more surprised by the conclusion that dopamine-transporter blockade, which is known to increase both tonic levels and acute releases, has very different effects on oICCS and eICSS: it increases the reinforcing intensity of oICCS but it does not increase the reinforcing intensity of eICSS. Because I am know the background that I summarized above, I was not as surprised by this as I assume most other readers will be. I think Shizgal knows that other readers in this area are strongly committed to the serial hypothesis, and that is why he spills so much ink trying to forestall objections to his conclusion that it cannot be true. Shizgal should remember Plank's often cited dictum that science progresses one funeral at a time. The conclusion that the serial hypothesis cannot be true is the single most featured conclusion in this paper, and justly so. But even this lead is more or less buried; it is not stated really plainly until far into the Discussion, after a great deal of highly technical detail aimed at forestalling possible objections to it (in the form of complex alternative explanations rooted in the bad frequency following of dopaminergic axons driven by trains of optical pulses and other such considerations). I strongly urge the author to move this lead forwards, and state it repeatedly, and in the simplest possible language, rather than burying it and putting it in technical language. It should be stated clearly in the Abstract and again in the Author Summary. It is already sort of there in both, but stated too technically (as strengthening the input to the signal integration process).

I was not clear what motivated the common reward mountain in the convergence model. Could not the locus of convergence be memory itself, not the integration of cost and intensity? Might not the trade-off between cost and reinforcing intensity be computed post memory at the time of deciding on an action? And need the trade-off be the same for qualitatively different intensities?

Bottom line: This is an unusually important paper, strongly challenging a widely and strongly held belief using state-of-the-art methods and exceptionally punctilious quantitative modeling of hypotheses specified in unusual detail.

Suggestions:

• In addition to making the data available at https://spectrum.library.concordia.ca/ , I suggest that they and the code used to obtain the results and in the modeling be uploaded to a public GitHub repository. Obviously, the simulation script should also go there. I have had more than one bad experience with supplementary materials supposedly preserved by journal on line. More than one of my own such depositions have become inaccessible. GitHub is much more trustworthy.

• p. 5: "This dependence is described by a surface in a three-dimensional space 140 (reward-seeking performance versus pulse frequency and response cost (Fig 1))." I suggest adding, closely analogous in its construction and interpretive function to the copula (the bivariate cumulative distribution function) in bivariate statistic. Although the audience for this paper includes everyone interested in the neural basis of reinforcement and the decision making that is driven by reinforcement (a broad audience), a particularly interested core constituency are computational neuroscientists, most of whom will be familiar with the central role copulas play in the analysis of correlated bivariate random variables. The reward-mountain figure looks exactly like a copula. This is no accident, because at the plateau at the top of the reward mountain represents describes the stimulation- and task-parameter space such that the probability of working for reinforcement is 1. The plateau of a copula describes the parameter space within which both cumulative probabilities are 1. Moreover, the function of copulas is to remove ambiguities inherent in the univariate cumulative probability functions. (A copula is a joint cumulative distribution function.)

• p.2: "Whereas the crude response counts used initially to 23 measure behavioral output are confounded by inherent non-linearity..." [should read ..."the inherent non-linearity of the performance function linking rate of response to reward magnitude, as well as ... because it is not clear to what function the mentioned nonlinearity applies.

• p. 9: I suggest that in the paragraph explaining why the results in Figure 4 are ambiguous, but that this ambiguity is removed by the reward-mountain method, which integrates the effects of the two variables (opportunity cost aka price=how long it has to work for each reward), you reveal the result of the disambiguation in the plainest possible language. I, for one, was keen to know which was the case and sorry that I had to wade through a lot of technical detail before finding out. More generally I suggest that the conclusions to be drawn from the shifts of the mountain observed in these experiments be stated plainly when these results are first mentioned. As it is, plain statements of the startling implications only appear much further on in the Discussion, after a great deal of technical detail. For most readers, the fact that the reward-mountain method generalizes to optical stimulation is a technical detail. I would have been surprised if it did not. Why should it not? The devil is always in the details, and no one pays closer attention to those details than Shizgal, which is why work from his lab is so trusted, but I urge that conclusions come first. In the present exposition, they are too often buried in details.

• I also suggest putting the discussion of other parametric work consistent with these results and conclusions be put after these conclusions from the present experiment in the Discussion rather than before, because the methods used here are so much more powerful than the methods used heretofore. There is altogether too much burying of the lead in this exposition. Those without a taste for technical detail may never get to the astonishing conclusions.

Reviewer #2: This paper from the Shizgal lab applies the group’s “reward mountain” concept to optical stimulation of ventral tegmental area (VTA) dopamine neurons. To derive the “reward mountain,” rats are given a series of trials in which two parameters are systematically varied: the price required to be paid to earn a reward (cumulative time spent holding down a lever), and reward magnitude (in this case, frequency of optical stimulation of dopamine neurons). The Shizgal lab has established previously that when the reward is electrical stimulation of the medial forebrain bundle (MFB), blocking dopamine reuptake shifts the mountain along the price axis, showing that drug-treated rats tended to be willing to pay a higher price for a given reward magnitude. At the same time, the drug caused very little if any movement along the “pulse frequency” axis, indicating that blocking dopamine reuptake does not alter the reward intensity of MFB stimulation at a given price point.

Given the above results, one might expect that blocking dopamine reuptake would have similar effects when the reward is optical stimulation of dopamine neurons. Surprisingly, however, administration of GBR12909 shifted the mountain along both axes – animals were both willing to pay a higher price, and the reward intensity of a given stimulation train tended to be higher. At first glance, this result is perplexing because MFB stimulation activates VTA dopamine neurons as well as directly activating other neuronal substrates that do not activate dopamine neurons. In contrast, optical stimulation presumably activates a more limited set of substrates (i.e., only dopamine neurons) – yet GBR12909 has an additional effect on the reward mountain that does not occur when MFB stimulation is used. To solve this apparent paradox, the authors devise a model in which the dopamine system is not the final common path that governs reward intensity. Rather, dopamine neurons contribute to reward intensity in parallel with other substrates that are activated by MFB stimulation. The authors argue (based on prior studies) that the directly-stimulated MFB axons that activate dopamine neurons cannot follow high-frequency stimulation, and therefore the level of dopamine neuron activation resulting from MFB stimulation is limited compared with that attainable from optical stimulation alone. From modeling the transformation between stimulation frequency and reward intensity (Fig. 11), the authors conclude that the upper limit to which dopamine reuptake blockade can increase reward intensity is greater for optical stimulation of dopamine neurons than for electrical MFB stimulation.

This result is very important because it addresses the long-standing controversies surrounding dopamine’s behavioral function (i.e., whether it plays role in motivation, effort exertion, reinforcement, learning, or some combination of these). The result forces us to re-think the interpretation of 50 years of experimentation with electrical MFB stimulation with respect to dopamine, and additionally forces us to broaden our thinking about what optogenetic stimulation of dopamine neurons actually does at behavioral, circuit and computational levels. In this regard, the authors include a section of the Discussion devoted to the vigorous promotion of careful mathematical modeling in concert with interventional behavioral experimentation – which is far superior to intuitive “X increases Y” neurobiological explanations of behavior. This advertisement for the future of neuroscience, coupled with the authors’ wry turns of phrase and their startling finding, is worth the hefty price of admission (with many factors on the price axis, the presence of 49 main and supplemental figures being just one).

One argument the authors anticipate is that because their conclusions rest so heavily on computational modeling, those conclusions are susceptible to erroneous assumptions and incorrect parameters. I am sympathetic to their defense of their approach: if it is to be criticized, it should be criticized on specific grounds (e.g., “parameter A is incorrect as shown by such-and-such study”) and not on general grounds (“you can conclude anything with enough equations”). At present, I have no specific grounds for criticism.

I do, however, have a few suggestions:

1. The argument developed to explain why the parallel model accommodates the GBR12909-induced shift along the pulse frequency axis refers to “tonic dopamine” (in the Fig. 11 legend). But the authors nowhere explicitly state what they think the roles of tonic and phasic dopamine are with respect to the two mountain axes. Such an explanation would be helpful.

2. On line 483, the authors write, “the rat is willing to sacrifice more leisure in order to obtain strong stimulation of a "good" eICSS site than a poorer one.” As the price gets higher, could it be that the animal is not so much interrupting effort to indulge in leisure, but rather he becomes fatigued, and a higher reward intensity makes it worth his while to work despite his fatigue? One could argue that taking a break from work is simply leisure, but the point here is that higher work requirements could induce fatigue and lead to more leisure-seeking.

3. Somewhat related to point #2, a growing literature implicates VTA dopamine neurons in arousal. To what extent could shifts along the price axis, such as that induced by GBR12909, be due to changes in arousal?

4. In 1905, a paper was published in Annalen der Physik entitled “On the Electrodynamics of Moving Bodies.” Despite its nondescript (and uninformative) title, the paper was widely read by physicists, who grew to appreciate the equations and ideas in the paper that described the Special Theory of Relativity. Alas, neuroscience is not like physics. A catchy title makes a big difference (e.g., “Accumbal D1R Neurons Projecting to Lateral Hypothalamus Authorize Feeding”). Fortunately, the authors have already supplied the beginnings of a title that actually describes their conclusions (unlike their present one) on line 980: “Dopamine neurons do not constitute an obligatory stage in the final common path for [reward] evaluation and pursuit.”

5. Some speculation about the identity of the MFB component that contributes to reward intensity in a dopamine-independent way would be helpful. It might also be useful to include a simplified version of figure 10 that explains in intuitive terms why the GBR12909 causes shifts along the reward intensity axis for optical stimulation but not electrical stimulation reward.

6. PLOS authors have the option to publish the peer review history of their article (what does this mean?). If published, this will include your full peer review and any attached files.

Reviewer #1: Yes: Charles R Gallistel

Reviewer #2: No

---

## [Author Response · Author response to Decision Letter 0]

16 Apr 2020

Please see the uploaded file: PLOS_ONE_rebuttal_letter_v3.pdf

---

## [Editor Report · Decision Letter 1]

6 May 2020

Dopamine neurons do not constitute an obligatory stage in the final common path for the evaluation and pursuit of brain stimulation reward

PONE-D-19-33493R1

Dear Dr. Shizgal,

I am pleased to inform you that your manuscript has been judged scientifically suitable for publication and will be formally accepted for publication once it complies with all outstanding technical requirements. 

With kind regards,

Juan M Dominguez, PhD

Academic Editor

PLOS ONE

Additional Editor Comments (optional):

I appreciate the effort that went into adequately responding to Reviewer's suggestions and questions. These changes greatly improved the manuscript. 
---

## [Editor Report · Acceptance letter]

21 May 2020

PONE-D-19-33493R1 

Dopamine neurons do not constitute an obligatory stage in the final common path for the evaluation and pursuit of brain stimulation reward 

Dear Dr. Shizgal:

I am pleased to inform you that your manuscript has been deemed suitable for publication in PLOS ONE. Congratulations! Your manuscript is now with our production department. 

With kind regards,

on behalf of

Dr Juan M Dominguez 

Academic Editor

PLOS ONE